# Structure Abstraction and Generalization in a Hippocampal-Entorhinal Inspired World Model

**Tianqiu Zhang** [* 1] **Muyang Lyu** [* 1] **Xiao Liu** [2] **Si Wu** [1]

hpc-mec-worldmodel.github.io

## Abstract

Humans abstract experiences into structured representations to facilitate pattern inference and knowledge transfer. While the hippocampal-entorhinal (HPC-MEC) circuit is known to represent both spatial and conceptual spaces, the mechanisms for concurrently extracting abstract structures from continuous, high-dimensional dynamics remain poorly understood. We propose a brain-inspired hierarchical model that simultaneously infers latent transitions and constructs a predictive visual world model. Our architecture employs an inverse model for structural extraction alongside an HPC-MEC coupling model that dissociates relational structures (MEC) from integrated episodic scenes (HPC). Using primitive transformation dynamics as a benchmark, we demonstrate the model's capacity for structural abstraction. By leveraging velocity-driven path integration, the framework enables robust prediction and structural reuse across diverse contexts, thereby achieving structural generalization. This work provides a novel computational framework for understanding how brain-inspired, self-supervised learning of world models facilitates the acquisition of reusable abstract knowledge.

## 1. Introduction

The hippocampal-entorhinal circuit has traditionally been studied in the context of spatial memory and navigation (Keefe & Nadel, 1978; Hafting et al., 2005). Re-

[1]Peking-Tsinghua Center for Life Sciences, Academy for Advanced Interdisciplinary Studies, IDG/McGovern Institute for Brain Research, Center of Quantitative Biology, School of Psychological and Cognitive Sciences, Key Laboratory of Machine Perception (Ministry of Education), Peking University. [2]HHMI Janelia. Correspondence to: Si Wu <siwu@pku.edu.cn>.

*Proceedings of the 43rd International Conference on Machine Learning*, Seoul, South Korea. PMLR 306, 2026. Copyright 2026 by the author(s).

cently emerging research demonstrates that this circuit extends beyond physical space navigation to encode abstract conceptual spaces, supporting diverse high-level cognitive tasks (Behrens et al., 2018). This neural architecture provides cognitive scaffolds for understanding abstract relationships by factorizing content and structure representations (Lerousseau & Summerfield, 2024). Substantial experimental and theoretical evidence (Manns & Eichenbaum, 2006; Whittington et al., 2020) supports a functional division: the hippocampus (HPC) binds content-specific information from individual experiences (Eichenbaum, 2017), while the medial entorhinal cortex (MEC) encodes abstract structures (Julian et al., 2018; Bao et al., 2019). This separation enables structural generalization, allowing the system to bind extracted structural representations flexibly with novel contexts (Kemp & Tenenbaum, 2008).

Grid cells within the MEC are fundamental to constructing these abstract structures (Behrens et al., 2018). Characterized by their periodic hexagonal firing patterns at various spatial scales (Giocomo et al., 2011), grid cells with the same spacing are organized into modules that function as continuous attractor neural networks (CANNs) (Amari, 1977; Ben-Yishai et al., 1995; Wu et al., 2008). Furthermore, when receiving velocity inputs, grid cells drive network activity across this attractor manifold, enabling path integration in abstract spaces (Burak & Fiete, 2009; Gardner et al., 2022) and facilitating mental simulation and planning.

The synergy between the HPC and MEC enables the binding of context-specific content to abstract relational structures (Whittington et al., 2020; Chandra et al., 2025). By performing path integration within the MEC to predict subsequent states, this circuit effectively serves as a biological implementation of a world model (Ha & Schmidhuber, 2018; LeCun, 2022). Within this framework, the abstract structures maintained by the MEC parallel the concept of latent actions in policy learning (Bruce et al., 2024; Schmidt & Jiang, 2024), where transitions are encoded into compact representations based on their underlying dynamics.

Despite these conceptual alignments, existing cognitive map models in neuroscience often struggle to scale, failing to demonstrate how such abstract structures can be extracted

directly from high-dimensional, real-world visual scenes. To bridge this gap, we develop a brain-inspired world model that abstracts the principal functions of the HPC-MEC circuit to learn complex transition dynamics from raw video sequences. Our research addresses two fundamental questions:

- *How can a model simultaneously learn representations of concrete sensory content and extract abstract structures from sequences without prior supervision?*

- *How can these extracted structures be leveraged to facilitate robust structural generalization across diverse objects and environments?*

In this work, we propose a hierarchical world model inspired by the HPC-MEC circuit capable of concurrently inferring abstract structures and learning a meaningful latent space from real-world sequences. The model comprises two components: an inverse model that extracts abstract latent transitions from sequences (Section 3.2), and an HPC-MEC-inspired hierarchical world model that extracts abstract structures from sequences while learning to predict the next frame through velocity-driven path integration (Section 3.1). Our model demonstrates robust capabilities of flexibly reusing abstract structures across different environments and objects. Furthermore, it exhibits effective predictive performance in real-world human activity scenarios and generalizes to previously unseen environments (Section 5).

The key contributions of this work are as follows:

- **Self-supervised learning of HPC-MEC world model for structure abstraction:** We introduce a self-supervised framework that jointly infers abstract structures and learns a HPC-MEC-inspired world model. The input consists solely of observation sequences, without requiring explicit priors on the dynamics.

- **Analysis of structure abstraction:** We use primitive transformation dynamics as a controllable example to show that the model decouples appearance from dynamics, demonstrating periodicity and in-class structure sharing.

- **Structure generalization across different contexts:** We propose a brain-inspired hierarchical world model that learns to extract shared structures from similar transition dynamics and flexibly reuses abstract structures across varied environments and object categories. This transfer capacity is enabled by the integration of abstract structures encoded in the MEC with content details stored in the HPC. Our model also shows generalization capabilities to out-of-distribution datasets.

## 2. Related works

**Cognitive map models.** Cognitive map models typically attribute structural abstraction to MEC, sensory binding to HPC, and sensory prediction to their interaction. The Tolman-Eichenbaum Machine (TEM) (Whittington et al., 2020) extends beyond spatial domains by using recurrent networks to form cognitive maps that predict observations from predefined actions, though it requires re-learning abstract maps across environments. Clone-structured cognitive graphs (CSCG) (George et al., 2021) offer graph-based Markovian representations of structural relationships without prior constraints, but remain limited to discrete domains. Vector-HaSH (Chandra et al., 2025) generates velocity inputs from hippocampal states to drive grid cells forming episodic memories, but its velocity vectors are learned by memorizing the whole sequence. All these works lack the critical step of inferring shared abstract structure from sequences in continuous and real-world environments.

**Abstract velocity extraction.** In neuroscience, (Iyer et al., 2024) employs an inverse model to extract abstract velocities from high-dimensional observations and maps them to low-dimensional grid cell velocity inputs. However, this approach significantly simplifies the complex representation learning in HPC-MEC circuits, focusing only on simple artificial stimuli.

**Infer latent transitions from the observations.** World models (Ha & Schmidhuber, 2018; LeCun, 2022) are generative models that predict future observations based on past observations and actions. When ground-truth actions are unavailable, inverse and forward dynamics models are commonly employed to infer actions and predict future states from observation-only demonstrations (Bruce et al., 2024; Ye et al., 2022; Schmidt & Jiang, 2024). Genie (Bruce et al., 2024), FICC (Ye et al., 2022), and LAPO (Schmidt & Jiang, 2024) primarily target 2D gaming environments, while LAPA (Ye et al., 2024), Moto (Chen et al., 2024b), IGOR (Chen et al., 2024a), and UniVLA (Bu et al., 2025) focus on latent action extraction from real-world settings for pretraining policy models. AdaWorld (Gao et al., 2025) achieves latent action transfer but relies on a large-scale pretrained video diffusion model. While their framework utilizes AdaLN (Peebles & Xie, 2023) for state-action interactions, our model leverages biologically-grounded CANN dynamics to achieve this integration.

## 3. Methods

The model is mainly separated into two parts: the HPC-MEC coupling model (Fig. 1(A, B)) and the inverse model (Fig. 1(C)). First, we use the pretrained multi-scale VQ-VAE (Tian et al., 2024) to extract observation embeddings

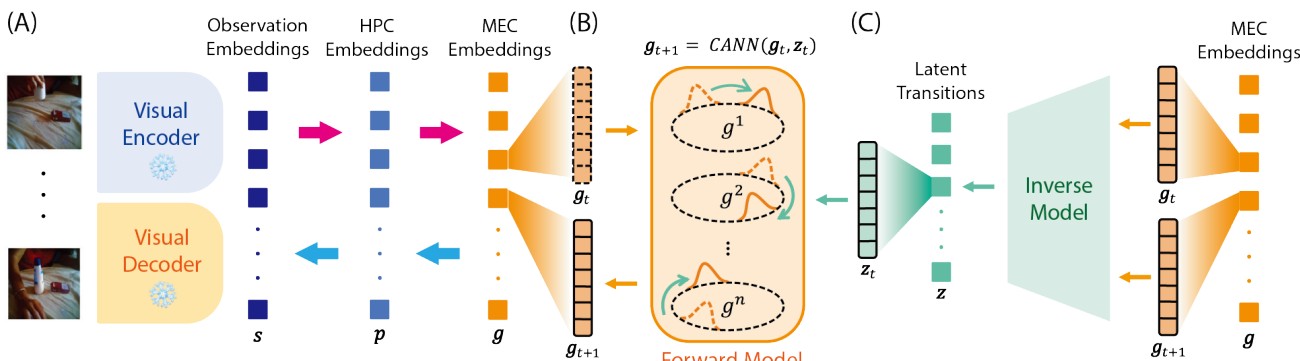

*Figure 1.* **Overview of the model architecture.** (A) Video clips are passed through the visual encoder to obtain observation embeddings $s$, which are encoded through the HPC to produce HPC embeddings $p$, and then passed to the MEC to generate MEC embeddings $g$. Finally, the generative pathway decodes them into the observation. The multi-scale VAE is fixed during training. (B) The latent transition $z_t$ operates on the MEC embedding $g_t$ at time $t$ using continuous attractor dynamics to generate the next MEC embedding $g_{t+1}$. (C) The inverse model is used to infer latent transitions from the MEC embeddings $g$.

from video sequences. The HPC-MEC coupling model hierarchically extracts abstract structures and performs next-state prediction via velocity-driven path integration (Fig. 1(B)). The inverse model is employed to infer latent transitions from the MEC embeddings, which encapsulate the underlying abstract structures. Finally, the decoder of VQ-VAE reconstructs the next frame from the generated observation embedding. The overall process is illustrated in Fig. 1.

### 3.1. The HPC-MEC-inspired Hierarchical World Model

The HPC-MEC coupling model is a hierarchical encoder-decoder architecture comprising two principal information flows: the visual inference flow and the generation flow. Rather than functioning solely as a reconstruction-based encoder–decoder, the model performs path integration by applying latent transitions to its MEC embeddings, enabling it to generate subsequent observations and thus serves as a world model. The graphical model of the HPC-MEC coupling model is illustrated in Fig. 2(A).

**The visual inference flow.** Input video frames $o_{1:T}$ are processed by the VQ-VAE encoder to obtain observation embeddings $s_{1:T}^{\text{inf}}$ (see Appendix A.3 for details). $s_{1:T}^{\text{inf}}$ are then encoded into higher-dimensional HPC embeddings $p_{1:T}^{\text{inf}}$ that capture the content information. Finally, the MEC compresses $p_{1:T}^{\text{inf}}$ into lower-dimensional MEC embeddings $g_{1:T}^{\text{inf}}$. Both the HPC and MEC use spatial-temporal Transformer encoders with temporal causal masking (Ye et al., 2024) to capture time dependencies.

**The generation flow.** The transition dynamic of the MEC is implemented as CANN-inspired template matching (Wu et al., 2008; Yoon et al., 2013) capable of performing path integration. A Continuous Attractor Neural Network (CANN) is a specific type of recurrent neural network designed to maintain a stable, continuous representation of information

in its activity pattern. The CANN maintains the current structural state as a stable, localized "bump" of neural activity within its metric space. This inherent geometric regularity is what makes them well-suited for path integration of grid cells (Burak & Fiete, 2009; Gardner et al., 2022). The key is that the network's translation-invariant geometric structure facilitates flexible state transitions through operators (see Appendix A.4 for mathematical details). In our model, the inferred latent transition $z_t$ serves as precisely such an operator. The latent transition controllably shifts the activity bump along the network's metric axes. To simplify the CANN dynamics, each dimension of the MEC embedding $g_t$ is represented by the bump center of a one-dimensional CANN: $g_t = [g_t^1, g_t^2, \ldots, g_t^n]$, where $n$ is the number of CANNs and $g_t^i$ is the bump center of the $i$-th CANN at time $t$ (Fig. 2(B)). Then $z_t$ is transformed into a concrete velocity term through its integration with the MEC embedding $g_t$. Specifically, the model first maps the concatenation of $z_t$ and $g_t^{\text{gen}}$ to produce a displacement vector $\Delta g_t^{\text{gen}}$, which serves as a direct velocity input to the CANN dynamics. The path integration dynamic can be simplified as the following equation at one discrete time step:

$$g_{t+1}^{\text{gen}} = g_t^{\text{gen}} \oplus \Delta g_t^{\text{gen}} = g_t^{\text{gen}} \oplus f_{\text{forward}}(z_t, g_t^{\text{gen}}), \quad (1)$$

where $\oplus$ represents the phase shift operator and $f_{\text{forward}}$ is implemented as a MLP. This update rule allows each dimension of $g_t$ to shift according to the corresponding velocity component, collectively forming the next MEC embedding $g_{t+1}$. Given that the dimensionality of latent transition $z_t$ is smaller than that of $g_t$, $f_{\text{forward}}$ serves as a transformation that combines the latent transition with its corresponding MEC embedding to generate a displacement vector $\Delta g_t^{\text{gen}}$. Through this mechanism, the model can transform latent transitions into concrete dynamics to predict future states.

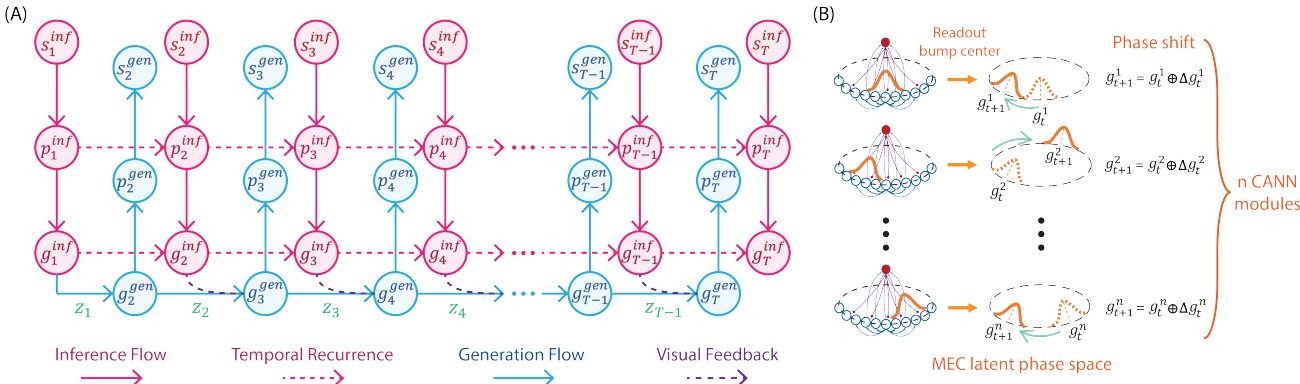

Inference Flow  Temporal Recurrence  Generation Flow  Visual Feedback

*Figure 2.* **Overview of the HPC-MEC coupling model.** (A) The graphical model of the HPC-MEC coupling model. The visual inference flow (solid pink arrow) models the encoding process of $\boldsymbol{s}_{1:T}^{\text{inf}} \to \boldsymbol{p}_{1:T}^{\text{inf}} \to \boldsymbol{g}_{1:T}^{\text{inf}}$. The temporal dependence (dashed pink arrow) ensures the continuity and consistency of the representations. The generation flow (solid blue arrow) models the transition dynamic and the decoding process of $\boldsymbol{g}_{2:T}^{\text{gen}} \to \boldsymbol{p}_{2:T}^{\text{gen}} \to \boldsymbol{s}_{2:T}^{\text{gen}}$. The visual feedback (dashed purple arrow) can correct the accumulated path integration error. (B) The mechanism of velocity-like abstractions operating in the CANN-inspired MEC latent space.

**The visual feedback.** This model enables autoregressive prediction by providing an initial observation and a sequence of latent transitions. Analogous to grid cells in the MEC performing path integration using velocity inputs, our model composes latent transitions to predict future states. However, solely relying on path integration inevitably accumulates errors over time. The visual inference flow addresses this by providing feedback to correct the accumulated errors. Specifically, when visual input is available, the model corrects the current state using the inferred MEC embedding $\boldsymbol{g}_t^{\text{inf}}$ from the observation to predict the next state:

$$\boldsymbol{g}_{t+1}^{\text{gen}} = \boldsymbol{g}_t^{\text{inf}} \oplus f_{\text{forward}}(\boldsymbol{z}_t, \boldsymbol{g}_t^{\text{inf}}). \qquad (2)$$

**The hierarchical separation of abstract structures.** In contrast to earlier world models (Schmidt & Jiang, 2024; Ye et al., 2024; Chen et al., 2024b) that integrate transition dynamics and reconstruction within a single latent space, our HPC-MEC coupling model distinctly encodes content-rich dynamics and underlying abstract structures. The MEC focuses on learning abstract structures of transition dynamics, whereas the HPC maintains richer contextual information for reconstruction. This separation promotes feature reuse and efficient representation: the MEC captures shared dynamics across objects, while the HPC retains the time-dependent episodic memory. As a result, the model generalizes transition patterns across visual contexts without conflating appearance with abstract dynamics.

### 3.2. Inferring Latent Transitions

Rather than directly mapping from the high-dimensional observation space, the inverse model infers the latent transition $\boldsymbol{z}_t$ from consecutive MEC embeddings $\boldsymbol{g}_t^{\text{inf}}$ and $\boldsymbol{g}_{t+1}^{\text{inf}}$. In the absence of self-motion, transitions between embeddings serve as a practical proxy for learning latent transitions. We build on this by framing latent transitions as movements

on a cognitive map. Our task is thus to learn these abstract latent transitions from state differences to derive an abstract cognitive map. The transitions between MEC embeddings capture the most salient and noise-free dynamics, which the inverse model then distills into a low-dimensional latent transition space:

$$\boldsymbol{z}_t = f_{\text{inverse}}(\Delta \boldsymbol{g}_t^{\text{inf}}) = f_{\text{inverse}}(\boldsymbol{g}_{t+1}^{\text{inf}} \ominus \boldsymbol{g}_t^{\text{inf}}), \qquad (3)$$

where $\ominus$ represents the phase difference operator and $f_{\text{inverse}}$ is implemented as a MLP. By operating on the difference between sequential MEC embeddings, $\boldsymbol{z}_t$ captures only the low-dimensional transition dynamics while excluding redundant visual features already encoded in the HPC.

Therefore, the inverse dynamics model $f_{\text{inverse}}$ compresses $\Delta \boldsymbol{g}_t^{\text{inf}}$ into $\boldsymbol{z}_t$, and the forward dynamics model $f_{\text{forward}}$ utilizes $\boldsymbol{z}_t$ alongside $\boldsymbol{g}_t^{\text{gen}}$ to predict $\Delta \boldsymbol{g}_t^{\text{gen}}$ to derive the next MEC embedding $\boldsymbol{g}_{t+1}^{\text{gen}}$ (Equation 1). In this way, the model operates akin to a transition $\Delta \boldsymbol{g}_t$ autoencoder optimized via a path integration task, which further encourages latent transition $\boldsymbol{z}_t$ to capture compressed abstract structures.

### 3.3. Training Stages

We train the inverse model and the HPC-MEC coupling model in a self-supervised learning paradigm. Our input consists solely of video sequences, thereby eliminating the need for any prior constraints on the underlying dynamics. We employ an alignment loss, adapted from (Whittington et al., 2020), to enforce consistency between sensory input and self-motion cues, mimicking the stable spatial coding of the HPC-MEC system. We divide the training process into three stages:

1. **Training the visual inference flow and the decoding process of the generation flow:** This stage trains the model to reconstruct observation embeddings to form

meaningful HPC embeddings $p_{1:T}^{\text{inf}}$ and MEC embeddings $g_{1:T}^{\text{inf}}$. The training objective combines reconstruction, alignment, and regularization losses:

$$
\begin{aligned}
\mathcal{L}_{\text{phase1}} = {} & \mathcal{L}_{\text{reconstruction}}(s_{1:T}^{\text{inf}}, s_{1:T}^{\text{rec}}) \\
& + \mathcal{L}_{\text{alignment}}(p_{1:T}^{\text{inf}}, p_{1:T}^{\text{rec}}) \\
& + \mathcal{L}_{\text{regularization}}(p_{1:T}^{\text{inf}}, g_{1:T}^{\text{inf}}),
\end{aligned}
\tag{4}
$$

where $s_{1:T}^{\text{rec}}$ and $p_{1:T}^{\text{rec}}$ are the reconstructed observations and HPC embeddings from $g_{1:T}^{\text{inf}}$ respectively. The reconstruction loss and the alignment loss are measured using MSE. The regularization loss encourages the model to learn a structured latent space by minimizing the covariance and variance of the embeddings (Bardes et al., 2022).

2. **Training the inverse model and the transition dynamics of the generation flow:** Using the meaningful embeddings obtained in stage 1, we train the inverse model to infer latent transitions $z$ from consecutive MEC embeddings $g^{\text{inf}}$. Simultaneously, we train the transition dynamics to predict the generated MEC embedding $g^{\text{gen}}$. We focus on one-step prediction to avoid accumulated errors from multi-step path integration and prevent the model from learning shortcuts that bypass latent transitions. The training objective extends phase 1 with additional alignment and transition losses:

$$
\begin{aligned}
\mathcal{L}_{\text{phase2}} = {} & \mathcal{L}_{\text{phase1}} \\
& + \mathcal{L}_{\text{alignment}}(g_{2:T}^{\text{inf}}, g_{2:T}^{\text{gen}}) \\
& + \mathcal{L}(f_{\text{fwd}}(z_{1:T-1}, g_{1:T-1}^{\text{inf}}), \Delta g_{1:T-1}^{\text{inf}}).
\end{aligned}
\tag{5}
$$

3. **Jointly finetuning the HPC-MEC coupling model and the inverse model:** In this final stage, we jointly finetune all model parameters using the phase 2 loss function. Unlike phase 2, the model now learns to autoregressively generate rollouts by performing path integration and forward prediction, using only the $g_1^{\text{inf}}$ and the latent transition sequence from the inverse model. The MEC embeddings learn to capture dynamics-relevant abstract structures at this stage. Detailed descriptions of model training in different stages are discussed in Appendix B.1.

### 3.4. Experimental Setup.

We evaluate our model using three types of datasets. The model is trained on Something-Something v2 (SSv2) dataset (Goyal et al., 2017), a large-scale human activity video dataset including complex interactions with objects without explicit action labels. Several simulated datasets are used to evaluate the out-of-distribution capability of our model, including 3D object transition datasets (COIL-100 (Nene et al., 1996), MIRO (Kanezaki et al., 2018), *OmniObject3D* (Wu et al., 2023)) and simulated environments

(Franka Kitchen (Gupta et al., 2019), Block Pushing (Florence et al., 2022), Push-T (Chi et al., 2023), and LIBERO Goal (Liu et al., 2023).)

## 4. Analysis of Structure Abstraction

As detailed in Section 5, our model achieves strong one-step and autoregressive predictions, alongside robust structural generalization across contexts in both complex human-object interactions (SSv2) and diverse OOD datasets. To uncover the representations driving these capabilities and validate the model's capacity to extract abstract structures from sequential data, we evaluate our SSv2-pretrained model on an OOD 3D object dataset rendered from OmniObject3D, featuring controlled transitions in rotation, translation, and scaling. We specifically analyze how hierarchical processing between HPC and MEC embeddings facilitates the emergence of shared structural representations. Finally, we extend this analysis to demonstrate that such structural abstraction readily generalizes to robotic simulated environments.

### 4.1. Qualitative Analysis

By visualizing both HPC and MEC latent spaces of 3D rotating objects, we demonstrate the model's ability to dissociate specific object content from underlying abstract dynamics.

**Periodic shared structures.** We first identify objects with periodic rotation patterns and categorize them into three periodicity classes based on visual similarity: period 1 (360°), $\frac{1}{2}$ (180°), and $\frac{1}{4}$ (90°). Through dimensional reduction analysis using UMAP (McInnes et al., 2018), we observe that these periodicity classes exhibit distinctive low-dimensional trajectories in the embedding space (Fig. 3(A)). The object with period 1 forms a complete circular trajectory, while the object with period $\frac{1}{2}$ forms two overlapping circular trajectories. The object with period $\frac{1}{4}$, with three white sides and one brown side, reuses the latent transition structure for consecutive similar visual transitions, forming two overlapping small circular trajectories, with the remaining distinct transitions forming a larger half-period circular trajectory. Both HPC and MEC embeddings exhibit similar periodicity patterns, but MEC representations form more clearly defined shared rotation features. To verify this, we perform multi-class analyses as follows.

**In-class shared structures.** To analyze in-class shared structures, we examine three object categories: pumpkins, red apples, and yellow apples, each containing multiple instances. We process rotation sequences of each object using our model to extract HPC and MEC embeddings, and visualize these representations with UMAP. The results (Fig. 3(B)) demonstrate that while both HPC and MEC exhibit inter-category separation, $p^{\text{inf}}$ additionally shows clear intra-category differentiation. Specifically, for differ-

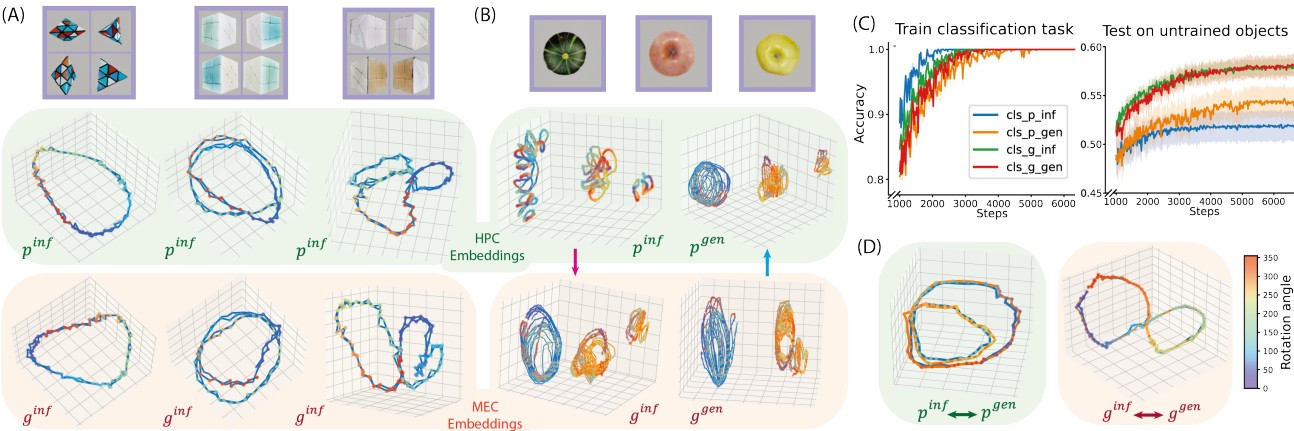

*Figure 3.* **Analysis of HPC and MEC embeddings.** (A) UMAP visualization of HPC and MEC embeddings grouped by periodicity class. Each object completes two full rotations. (B) UMAP visualization of HPC and MEC embeddings grouped by object category. (C) Classification accuracy of object categories using HPC and MEC embeddings. (D) Alignment between inference and generation embeddings for an individual object.

ent in-class objects, $p^{\text{inf}}$ forms separated circular trajectories, reflecting object-wise features. In contrast, $g^{\text{inf}}$ and $g^{\text{gen}}$ trajectories substantially overlap, capturing the in-class shared rotational features. These shared structures propagate through the generation flow, shape the manifold of $p^{\text{gen}}$, making rotational features more salient in the high-dimensional space. Furthermore, while the global manifolds of $p^{\text{inf}}$ and $p^{\text{gen}}$ naturally exhibit differences, UMAP projections of individual objects' inference and generation embeddings maintain high consistency in both HPC and MEC (Fig. 3(D)).

### 4.2. Quantitative Analysis

We also conduct some quantitative experiments to demonstrate that HPC binds more specific content information, while MEC encodes more generalizable abstract structures.

**Quantification of in-class structural sharing.** To quantify that the MEC abstracts in-class shared structures while the HPC holds more identity features (stated in Section 4.1), we train separate lightweight decoders to classify object categories from HPC and MEC embeddings. Test objects are excluded during training, so higher test accuracy indicates greater structural consistency within object classes. Our results show that after training convergence, the test accuracy on $p^{\text{inf}}$ is consistently lower than $g^{\text{inf}}$ and $g^{\text{gen}}$, indicating that HPC embeddings encode more content-specific details. In contrast, MEC embeddings better capture in-class shared structures, enhancing generalization to novel instances (Fig. 3(C)).

**Transition decoding experiment.** We use 3D-object transition sequences containing different transformations of rotation, translation, and scaling. Each sequence features an object with randomized category, initial position, orienta-

tion, and size. We feed these sequences to our model for zero-shot generalization and train transition decoders of a uniform hidden size on the resulting latents to predict the transition type. This allows us to measure the abstraction and transition semantics of the latent representations. From the results shown in Table 1, we can find that the latent transition ($z$) achieves the highest accuracy in decoding transition types, followed by state transitions in the MEC space ($\Delta g^{inf}$). While it is more difficult to abstract transition information from the HPC space ($\Delta p^{inf}$), because it contains more specific details. These results demonstrate that the MEC extracts more content-free structures that are highly effective and semantically meaningful.

*Table 1.* Decoding accuracy across different embeddings.

| Embedding | $p^{\text{inf}}$ | $g^{\text{inf}}$ | $\Delta p^{\text{inf}}$ | $\Delta g^{\text{inf}}$ | $z$ |
|---|---|---|---|---|---|
| Accuracy ↑ | $0.3330 \pm 0.0163$ | $0.3486 \pm 0.0156$ | $0.8386 \pm 0.0263$ | $0.8868 \pm 0.0212$ | $\mathbf{0.9064 \pm 0.0145}$ |

**Generalization to robotic dynamics.** To further validate our findings in complex environments, we analyze sequences of the same action (e.g., "opening the upper cabinet" in Franka Kitchen) performed under varying contexts (e.g., different object positions or microwave states). We then compute the cosine similarity of the state transitions within the HPC space ($\Delta p$), the MEC space ($\Delta g$), and the latent transition space ($z$). The results are shown in Table 2. We can see that latent transition trajectories exhibit the highest similarity, and the MEC space exhibits higher structural alignment than the HPC space. These results suggest that the model can effectively extract content-independent structures from robotic simulated environments.

*Table 2.* Sequence similarity of latent temporal differences.

| Transitions | $z$ | $\Delta g^{\text{inf}}$ | $\Delta g^{\text{gen}}$ | $\Delta p^{\text{inf}}$ | $\Delta p^{\text{gen}}$ |
|---|---|---|---|---|---|
| Cos Sim ↑ | $\mathbf{0.235 \pm 0.021}$ | $0.146 \pm 0.063$ | $0.152 \pm 0.057$ | $0.024 \pm 0.061$ | $0.114 \pm 0.056$ |

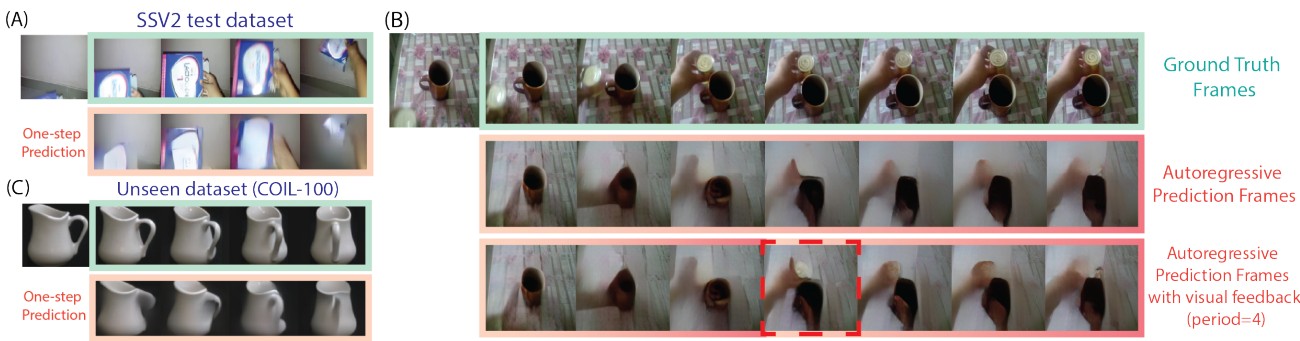

*Figure 4.* **Evaluation on one-step and autoregressive prediction.** (A) One-step prediction evaluated on the SSv2 test dataset. (B) Autoregressive prediction with and without the visual feedback on the SSv2 test dataset. (C) One-step prediction on an out-of-distribution dataset, COIL-100.

# 5. Episodic Synthesis and Structural Generalization

Having established the model's capacity for structural abstraction, we demonstrate that these latent structures serve a dual purpose: facilitating the synthesis of episodic memories and enabling robust structural generalization. Specifically, our framework leverages these abstract structures to transfer across novel entities. By decoupling the underlying transitions from specific sensory content, the model can generate analogous movements for entirely different objects or scenes, effectively demonstrating zero-shot transfer.

## 5.1. One-step and Autoregressive Predictions

For one-step prediction, the model successfully extracts latent transition from the input video and generates frames that match the dynamics of the ground-truth sequence (Fig. 4(A)). For autoregressive prediction, the model generates the entire sequence by applying a sequence of latent transitions to the initial frame (Fig. 4(B)). We find that the model maintains good consistency even when generating longer sequences, although the generation quality gradually decreases over time. This corresponds to the accumulating path integration errors in MEC. Similar to biological systems, the model can correct compound errors by receiving visual feedback from the sensory input. We test the model's performance after introducing visual feedback at the fourth rollout step and observe improved generation quality, with more accurate details in the subsequent frames (Fig. 4(B)). We conduct an additional ablation study to examine the role of latent transitions in governing transition dynamics, discussed in Appendix E.

**Generalization to out-of-distribution datasets.** To evaluate whether the model could effectively extract and utilize latent transitions to generate OOD scenes, we assess the model on unseen 3D object transition datasets. Our results show that the model successfully identifies fundamental rotational transformations as latent transitions and generates sequences that closely approximate the ground truth

(Fig. 4(C)). We further evaluate the model on simulated benchmarks with significant distributional shifts from human videos, potentially limiting generalization to virtual domains. Full results are provided in Appendix F. Our findings show the model performs more robustly in the more naturalistic environments like Franka Kitchen, but less effectively in artificial environments like Push-T.

## 5.2. Structural Generalization Across Contexts

To validate our model's ability to reuse latent transitions, we conduct evaluations using both naturalistic human activity data and simulated datasets.

The results of one-step latent transition reuse are shown in Fig. 5(A). We extract the latent transition from the purple-highlighted image pairs, which capture hand movements such as disappearing or grabbing objects. We then apply the same latent transition to different scenes and generate subsequent frames. The generated frames successfully mimic the dynamics of the original image pairs.

To further investigate the latent transition transferability, we conduct experiments applying latent transition sequences extracted from one video to frames containing different contexts. Using apple rotation as a case study, we extract latent transitions from a sequence featuring a yellow apple and apply them to generate the rotation of a ripe red apple. Our results reveal that while the texture features in the generated sequence correspond to the ripe red apple, the rotational dynamics align with the yellow apple from which the latent transitions are derived (Fig. 5(B)). When implementing autoregressive prediction using only the extracted latent transitions and the initial frame, we observe that the generated sequence maintains alignment with the rotational dynamics of the source sequence. However, the texture features progressively deviate, becoming more luminous than the ground truth sequence (Fig. 5(C)). We also present two examples to illustrate sequential latent transition reuse in human activity scenarios in Fig. 5(D,E). The resulting image sequence captures the dynamics from the source scenario

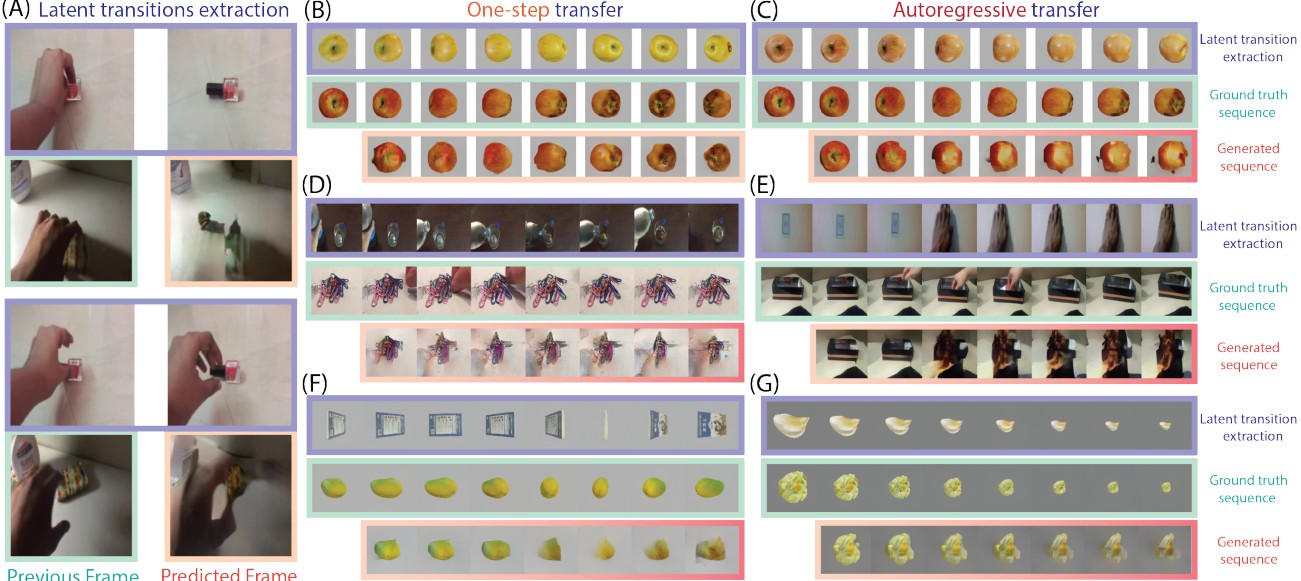

*Figure 5.* **Structural generalization.** (A) One-step latent transition transfer across different scenes on SSv2. (B)(C) One-step & autoregressive prediction by transferring the sequential latent transitions. (D)(E) Autoregressive reuse of latent transitions on SSv2. (F)(G) Autoregressive reuse of latent transitions on rotation and scaling dynamics across object categories.

while preserving the content information of the new scene. Fig. 5(F, G) demonstrates two additional cases: rotation and scaling dynamics transfer across object categories.

### 5.3. Baseline Comparison

We compare our model against state-of-the-art latent action models, LAPA (Ye et al., 2024), Moto (Chen et al., 2024b), and AdaWorld LAM (Gao et al., 2025), which align to extract and reuse latent transitions from observation-only videos without ground-truth action labels. We use two standard metrics: Structural Similarity Index (SSIM) (Wang et al., 2004) for local structural consistency and Learned Perceptual Image Patch Similarity (LPIPS) (Zhang et al., 2018) for perceptual similarity. As shown in Table 3, our model achieves lower LPIPS scores in both one-step and autoregressive prediction, indicating closer visual similarity with ground-truth dynamics and more effective extraction of latent transitions. On the Franka Kitchen dataset, both Moto and AdaWorld LAM struggle to predict meaningful dynamics, frequently degenerating into static predictions that are almost identical to the preceding frames. Conversely, our model maintains robust performance across both in-domain and OOD datasets. We attribute this to our HPC-MEC inspired architecture: unlike baselines that learn latent transitions in an entangled unified space, our hierarchical model learns them within a more structured space, yielding abstract latent transitions that significantly mitigate autoregressive compounding errors and enhance OOD resilience.

### 5.4. Ablation Study

To isolate the components driving our performance, we evaluate ablated variants on the OOD structure reuse task

*Table 3.* Quantitative comparison of SSIM and LPIPS across models and downstream datasets. Each model performs an 8-step sequence generation. * indicates out-of-distribution datasets.

| Dataset | Model | SSIM ↑ | | LPIPS ↓ | |
| --- | --- | --- | --- | --- | --- |
| | | one-step | autoregression | one-step | autoregression |
| SSv2 | LAPA | $0.744_{\pm 0.022}$ | $0.659_{\pm 0.027}$ | $0.357_{\pm 0.017}$ | $0.448_{\pm 0.017}$ |
| | Moto | $0.668_{\pm 0.027}$ | $0.566_{\pm 0.029}$ | $0.301_{\pm 0.022}$ | $0.480_{\pm 0.012}$ |
| | AdaWorld(LAM) | $\mathbf{0.763_{\pm 0.023}}$ | $0.654_{\pm 0.018}$ | $0.295_{\pm 0.026}$ | $0.448_{\pm 0.022}$ |
| | VQ-VAE ablation | $0.717_{\pm 0.023}$ | $0.677_{\pm 0.031}$ | $0.313_{\pm 0.013}$ | $0.378_{\pm 0.017}$ |
| | Our model | $0.752_{\pm 0.019}$ | $\mathbf{0.687_{\pm 0.018}}$ | $\mathbf{0.274_{\pm 0.026}}$ | $\mathbf{0.356_{\pm 0.015}}$ |
| COIL-100* | LAPA | $0.589_{\pm 0.038}$ | $0.682_{\pm 0.020}$ | $0.301_{\pm 0.021}$ | $0.385_{\pm 0.016}$ |
| | Moto | $0.700_{\pm 0.031}$ | $0.642_{\pm 0.043}$ | $0.352_{\pm 0.041}$ | $0.473_{\pm 0.031}$ |
| | AdaWorld(LAM) | $0.778_{\pm 0.039}$ | $0.715_{\pm 0.042}$ | $0.312_{\pm 0.048}$ | $0.415_{\pm 0.060}$ |
| | Our model | $\mathbf{0.837_{\pm 0.021}}$ | $\mathbf{0.814_{\pm 0.032}}$ | $\mathbf{0.226_{\pm 0.034}}$ | $\mathbf{0.270_{\pm 0.031}}$ |
| Franka Kitchen* | LAPA | $0.690_{\pm 0.002}$ | $0.532_{\pm 0.003}$ | $0.389_{\pm 0.001}$ | $0.529_{\pm 0.001}$ |
| | Moto(failed) | - | - | - | - |
| | AdaWorld(LAM)(failed) | - | - | - | - |
| | VQ-VAE ablation | $0.576_{\pm 0.037}$ | $0.523_{\pm 0.038}$ | $0.385_{\pm 0.018}$ | $0.445_{\pm 0.022}$ |
| | Our model | $\mathbf{0.705_{\pm 0.004}}$ | $\mathbf{0.551_{\pm 0.005}}$ | $\mathbf{0.253_{\pm 0.003}}$ | $\mathbf{0.426_{\pm 0.003}}$ |

(Table 4). Successful reuse requires generated frames to align with the target sequence rather than the source sequence used to extract the latent transition. We quantify this by defining a similarity ratio using features from a pretrained DINOv2 encoder $E$. This ratio $R = \mathbb{E}\left[\frac{\cos(E(\mathbf{o}_t^{\text{gen}}), E(\mathbf{o}_t^{\text{content}}))}{\cos(E(\mathbf{o}_t^{\text{gen}}), E(\mathbf{o}_t^{\text{action}}))}\right]$ measures whether the generated object is closer to the target content or the source content. A higher R indicates stronger alignment with the content sequence and thus better latent transition reuse. We demonstrate the effectiveness of each core component using three different ablated variants:

**Hierarchical HPC–MEC separation.** We construct a unified space variant by removing the MEC layer to test the necessity of our disentangled architecture (denoted as "Our model w/ unified latent space"). By coupling the inverse model and CANN module directly to the single, high-dimensional HPC layer, the inverse model infers $\mathbf{z_t}$ from consecutive HPC embeddings ($\mathbf{p}_t^{\text{inf}}$, $\mathbf{p}_{t+1}^{\text{inf}}$). Conse-

*Table 4.* Quantitative comparison of ablation models on the OOD structure reuse task.

| Model | R ↑ | | SSIM ↑ | | LPIPS ↓ | |
|---|---|---|---|---|---|---|
| | one-step | autoregression | one-step | autoregression | one-step | autoregression |
| Our model w/ unified latent space | $2.054_{\pm 0.521}$ | $1.542_{\pm 0.246}$ | $0.901_{\pm 0.007}$ | $0.886_{\pm 0.008}$ | $0.126_{\pm 0.008}$ | $0.179_{\pm 0.008}$ |
| Our model w/o CANN | $2.403_{\pm 0.553}$ | $1.859_{\pm 0.396}$ | $0.894_{\pm 0.022}$ | $0.888_{\pm 0.009}$ | $0.149_{\pm 0.009}$ | $0.177_{\pm 0.010}$ |
| VQ-VAE ablation | $2.035_{\pm 0.229}$ | $1.796_{\pm 0.173}$ | $0.892_{\pm 0.009}$ | $0.883_{\pm 0.009}$ | $0.158_{\pm 0.009}$ | $0.177_{\pm 0.009}$ |
| Our model | $\mathbf{3.201_{\pm 0.435}}$ | $\mathbf{2.482_{\pm 0.460}}$ | $\mathbf{0.902_{\pm 0.010}}$ | $\mathbf{0.891_{\pm 0.009}}$ | $\mathbf{0.120_{\pm 0.008}}$ | $\mathbf{0.156_{\pm 0.008}}$ |

*Table 5.* Quantitative comparison of action decoding accuracy using latent transitions from different models.

| Model | AdaWorld(LAM) | LAPA | Moto | VQ-VAE ablation | Our model w/o CANN | Our model |
|---|---|---|---|---|---|---|
| Accuracy ↑ | $0.6395 \pm 0.0192$ | $0.8259 \pm 0.0155$ | $0.7190 \pm 0.0120$ | $0.8523 \pm 0.0247$ | $0.8768 \pm 0.0117$ | $\mathbf{0.9064 \pm 0.0145}$ |

quently, the CANN's path integration operates on these content-entangled states. This variant exhibits significant "texture leakage" from the source video, indicating that hierarchical separation is essential to isolate pure dynamics and prevent latent transitions from entangling with object-specific content.

**CANN-based MEC dynamics.** We ablate the MEC's internal dynamics by replacing our CANN module with a standard MLP of equivalent capacity ("Our model w/o CANN"). This variant performs "state-to-state" prediction, directly computing the next state from the concatenated current state and latent transition: $g_{t+1}^{\mathrm{gen}} = \mathrm{MLP}([g_t^{\mathrm{gen}}, z_t])$. In contrast, our CANN module predicts the relative transition $\Delta g_t$ (Section 3.2). This transition-centric design is crucial: it relieves $z_t$ from full state reconstruction, forcing it to encode purely content-independent dynamics. Consequently, the MLP modification fails completely in OOD transfer tasks, demonstrating the absolute necessity of the CANN structure for generalizing learned dynamics to novel scenes.

**Structure not from the pretrained encoder.** To confirm our abstract structures are driven by the HPC-MEC module rather than merely inherited from the visual encoder, we evaluate a "VQ-VAE ablation" lacking the hierarchical architecture entirely. This variant uses the identical pretrained VQ-VAE, an inverse model inferring $z_t$ directly from consecutive embeddings ($s_t^{\mathrm{inf}}, s_{t+1}^{\mathrm{inf}}$), and an MLP forward model predicting $s_{t+1}^{\mathrm{gen}} = \mathrm{MLP}([s_t^{\mathrm{gen}}, z_t])$. Without the HPC-MEC's structural abstraction, this variant retains excessive source information, yielding inferior transfer predictions (Table 4). This confirms that the robust extraction of content-free abstractions is fundamentally driven by our hierarchical design rather than the pretrained encoder.

### 5.5. Latent Transition Decoding

To evaluate the quality of latent transitions against baselines and ablations, we revisit the OOD 3D-object transition task (Section 4.2). Briefly, this involves sequences where diverse unseen objects undergo a single transformation (rotation, translation, or scaling). Following LAPO (Schmidt & Jiang, 2024), we train an MLP probe to classify these transfor-

mation types directly from the latent transitions produced by each model. As shown in Table 5, our model significantly outperforms all baselines and ablations. These results demonstrate that learning latent transitions in MEC space via the hierarchical model captures more abstract semantics than unified-space architectures (e.g., all baselines and the VQ-VAE ablation), which are prone to interference from content details. Moreover, the performance gap between our full model and the "w/o CANN" ablation highlights the critical role of the CANN module in suppressing content-specific details and yielding transitions that better reflect the underlying dynamics. This is consistent with our earlier finding in Section 4.2 that structured MEC-space transitions are more semantically decodable than HPC-space ones.

## 6. Discussion and limitations

Our finding shows how abstract structures can be effectively extracted from real-world video sequences while maintaining meaningful hierarchical latent spaces. The combination of the inverse model with the HPC-MEC-inspired world model enables efficient extracting abstract structures from specific contents, facilitating robust transfer capabilities. Our analysis of the HPC and MEC representations further highlights the potential for neuro-inspired models to encode and reuse abstract structures from real-world transition dynamics. The capability of our model to reuse these latent transitions across diverse contexts underscores the critical role of structural generalization.

**Limitations.** Our model has several limitations. First, it is susceptible to autoregressive compounding errors, which could potentially be mitigated by incorporating a memory bank. Second, performance degrades on benchmarks that exhibit substantial distributional drift from the training data (human videos). Additionally, coordinating multiple independent entities remains a major challenge. Future work will explore hierarchical HPC-MEC structures or object-centric representations to tackle this problem.

## Impact Statement

This paper presents work whose goal is to advance the field of Machine Learning. There are many potential societal consequences of our work, none of which we feel must be specifically highlighted here.

## Acknowledgements

This work was supported by the National Natural Science Foundation of China (no. T2421004 to S.W.), the National Key Research and Development Program of China (2024YFF1206500), the Science and Technology Innovation 2030-Brain Science and Brain-inspired Intelligence Project (no. 2021ZD0200204, S.W.).

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

# A. Model details

## A.1. Model design motivation

Our model is a brain-inspired framework guided by neuroscience and implemented with state-of-the-art deep learning modules.

The hierarchical separation of a content pathway (HPC) from a structure pathway (MEC) is directly inspired by established theories in computational neuroscience. Using a latent transition space learned via an inverse dynamics model is a common and effective framework adopted by many baselines in this field for learning from unlabeled video. Our work builds upon this solid foundation. Within this framework, our component choices are deliberate. The MEC's path integration role is implemented using a Continuous Attractor Neural Network (CANN), a classic computational model of grid cells (Gardner et al., 2022; McNaughton et al., 2006; Fuhs & Touretzky, 2006; Burgess et al., 2007). latent transitions are inferred via an inverse dynamics model, a standard approach rooted in theories of the cerebellum (Wolpert et al., 1998).

Finally, we choose specific state-of-the-art implementations for these functional roles to ensure high performance. The pretrained visual multi-scale VQ-VAE (Tian et al., 2024) provides a stable and high-quality visual representation, analogous to processed input from the visual cortex. The spatial-temporal Transformer (Bruce et al., 2024; Ye et al., 2024) models complex dependencies across space and time, exactly what is required to integrate the content and structure signals to predict future states.

## A.2. Model parameter

We provide the model parameters (Table 6) used in our experiments.

*Table 6.* Model parameters

| COMPONENT/PARAMETER | VALUE |
|---|---|
| **Input parameters** | |
| Input Channels | 3 |
| Input Image Height | 256 |
| Input Image Width | 256 |
| VQ-VAE Encoder Depth | 16 |
| VQ-VAE Encoder Feature Map Channels | 32 |
| VQ-VAE Encoder Feature Map Heights | 16 |
| VQ-VAE Encoder Feature Map Widths | 16 |
| Patch Size | 4 |
| Patch Height | 4 |
| Patch Width | 4 |
| | |
| **HPC Model** | |
| HPC Hidden Size (total) | 8192 |
| Per-patch Hidden Dimension | 512 |
| Spatial Transformer Depth | 4 |
| Temporal Transformer Depth | 4 |
| | |
| **MEC Model** | |
| MEC Hidden Size (total) | 4096 |
| Per-patch Hidden Dimension | 256 |
| Spatial Transformer Depth | 4 |
| Temporal Transformer Depth | 4 |
| | |
| **Inverse World Model** | |
| Transition Dimension | 2048 |
| Per-patch Transition Dimension | 128 |
| $f_{\text{inverse}}$ Residual Blocks | 2 |
| $f_{\text{forward}}$ Residual Blocks | 4 |

### A.3. Pretrained encoder details

We use the pretrained multi-scale VQ-VAE (depth=16) from the VAR model to extract visual features. The feature map we use, $\hat{f}$, is the sum of outputs from the multi-scale vector quantization layers. This feature map is fed directly into our model. This design is intentional: it simulates how the HPC-MEC circuit receives pre-processed information from the visual cortex, and it frames the task as a prediction problem entirely within the latent space (akin to JEPA (LeCun, 2022)), meaning our model does not perform pixel-level reconstruction.

### A.4. CANN Dynamics and Path Integration

Inspired by the role of grid cells in the MEC, we model transition dynamics using a continuous attractor neural network (CANN). CANN is a specific type of recurrent neural network designed to maintain a stable, continuous representation of information in its activity pattern. Unlike classical attractor networks that store discrete, unstructured patterns, CANNs are distinguished by their ability to encode structured patterns organized by metric relationships. This geometric structure allows a CANN to maintain a system's state as a stable, localized "activity bump". Path integration is achieved by using the inferred latent transition as a velocity operator that controllably shifts this bump along the network's metric axes. This ability to maintain a stable state and systematically transform it via a velocity input is what allows the CANN to integrate a path and predict the next state's structure.

In our model, the MEC embeddings $g$ represent abstract structures in high-dimensional space (Klukas et al., 2020). When multiple one-dimensional CANNs are combined, their joint state space forms an $N$-dimensional continuous manifold (Burgess & Burgess, 2014). This modular representation enables the decomposition of spatial dynamics, where each module is driven by latent transition inputs.

We formalize the CANN dynamics implemented in our MEC module for encoding abstract representations and performing path integration. Specifically, we draw connections between the classical differential equation-based formulation of CANNs and our discrete-time, learnable implementation.

The canonical dynamics of a one-dimensional CANN are described by the following first-order differential equation:

$$\tau \frac{d\mathbf{u}(t)}{dt} = -\mathbf{u}(t) + \mathbf{W}_r \mathbf{r}(t) + \mathbf{W}_{\text{in}} \mathbf{v}(t), \tag{6}$$

where:

- $\mathbf{u}(t)$ is the hidden state of the network;

- $\mathbf{r}(t)$ is the instantaneous neural firing rate, which is derived from the hidden state using a non-linear activation function $\mathbf{r}(t) = H(\mathbf{u}(t))$;

- $\mathbf{v}(t)$ is the external motion input (e.g., velocity);

- $\mathbf{W}_r$ and $\mathbf{W}_{\text{in}}$ are the recurrent and input weight matrices, respectively;

- $\tau$ is a time constant.

This formulation supports both self-sustaining activity through recurrent feedback and perturbation by external motion cues. To enable numerical modeling, Equation (6) is discretized using forward Euler integration:

$$\mathbf{u}_{t+1} = (1 - \alpha)\mathbf{u}_t + \alpha \left( \mathbf{W}_r \mathbf{r}_t + \mathbf{W}_{\text{in}} \mathbf{v}_t \right), \tag{7}$$

where $\alpha = \Delta t / \tau$.

For simplicity, we omit the explicit modeling of Gaussian bump profiles and instead use a nonlinear function to extract bump centers: $\mathbf{g}(t) = \sigma(\mathbf{r}(t))$. Rather than explicitly designing $\mathbf{W}_r$, we learn the continuous attractor dynamics through a temporal attention mechanism and impose structural regularity using a second-order temporal smoothing loss on $\mathbf{g}$. This loss penalizes high acceleration in the latent space and is given by:

$$\mathcal{L}_{\text{smooth}} = \frac{1}{B(T-2)} \sum_{b=1}^{B} \sum_{t=1}^{T-2} \|\mathbf{g}_{b,t+2} - 2\mathbf{g}_{b,t+1} + \mathbf{g}_{b,t}\|^2, \tag{8}$$

where $B$ is the batch size and $T$ is the sequence length. The temporal attention with causal masking integrates the historical information into the current state, encouraging low curvature in the grid trajectory, reflecting the physical intuition that continuous movement through space should induce smooth changes in internal state.

Focusing on the evolution of bump centers rather than firing rates, we implement the path integration update as:

$$\mathbf{g}_{t+1} = \mathbf{g}_t \oplus f(\mathbf{g}_t, \mathbf{z}_t), \tag{9}$$

where $\mathbf{z}_t$ denotes the latent transition and $f(\cdot)$ is a learnable function that models the dynamics induced by $\mathbf{z}_t$. The recurrent dynamics of $\mathbf{g}$ are embedded within the temporal attention structure to maintain continuity over time and to preserve the attractor manifold structure.

### A.5. Sensitivity analysis

For the number of CANN modules (MEC dimension), our current model uses 4096 modules (256 per visual patch). Reducing this to 2048 still allows the model to encode scene-specific dynamics, but with a noticeable loss of detail and increased blurriness in the generated images. A further reduction to 1024 exacerbates this issue and leads to convergence difficulties during training.

Regarding the latent transition dimension, we currently use a dimension of 2048. We find that the model can still predict the next frame with a dimension of 1024, though the generation quality is compromised. Compressing the latent transition dimension further makes convergence very difficult, often causing the model to learn a trivial solution where it simply outputs the previous frame as its prediction.

### A.6. Visual feedback details

Since path integration alone accumulates errors over time, the visual inference pathway provides corrective feedback to mitigate this drift. Specifically, in the absence of visual feedback, the model predicts the next state via path integration based on the current prediction $\boldsymbol{g}_t^{gen}$:

$$\boldsymbol{g}_{t+1}^{gen} = \boldsymbol{g}_t^{gen} \oplus f_{\text{forward}}(\boldsymbol{z}_t, \boldsymbol{g}_t^{gen}) \tag{10}$$

When visual input is available, the model corrects the current state using the inferred MEC embedding $\boldsymbol{g}_t^{inf}$ from the observation to predict the next state:

$$\boldsymbol{g}_{t+1}^{gen} = \boldsymbol{g}_t^{inf} \oplus f_{\text{forward}}(\boldsymbol{z}_t, \boldsymbol{g}_t^{inf}) \tag{11}$$

This update does not introduce instability because $\boldsymbol{g}_t^{inf}$ and $\boldsymbol{g}_t^{gen}$ lie in the same latent space. The alignment loss $\mathcal{L}_{\text{alignment}}(\boldsymbol{g}_{2:T}^{\text{inf}}, \boldsymbol{g}_{2:T}^{\text{gen}})$ not only trains the inverse model but also ensures that replacing $\boldsymbol{g}_t^{gen}$ with $\boldsymbol{g}_t^{inf}$ during feedback remains stable. Moreover, visual feedback is used only after the model has been sufficiently trained, at which point the two embeddings are well aligned. As a result, even though the model is not trained on long sequences, the correction mechanism allows it to autoregressively generate accurate predictions over time.

# B. Model training details

## B.1. Loss functions

We elaborate on the loss functions designed for each training phase in detail:

- **The reconstruction loss:** We include losses between observation embeddings and three different generative model predictions to accelerate learning. The reconstructed observation embeddings $s^{\text{recon}}$ can be generated directly from the inferred $p^{\text{inf}}$ or $g^{\text{inf}}$ embeddings, while $s^{\text{gen}}$ is generated from the generative pathway. In phase 1, we use $p_{1:T}^{\text{inf}} \to s_{1:T}^{\text{recon}}$ and $g_{1:T}^{\text{inf}} \to p_{1:T}^{\text{inf}} \to s_{1:T}^{\text{recon}}$ to compute the reconstruction loss. In phases 2 and 3, the transition dynamics of the model predict the next generated observation embeddings $g_{2:T}^{\text{gen}} \to p_{2:T}^{\text{gen}} \to s_{2:T}^{\text{gen}}$, and we use $s_{2:T}^{\text{gen}}$ to compute the reconstruction loss.

- **The alignment loss:** The $\mathcal{L}_{\text{alignment}}$ is used to align the latent representations from inference and generation. This loss is inspired by TEM (Whittington et al., 2020), where the inferred HPC embeddings $p^{\text{inf}}$ are aligned with the generated HPC embeddings $p^{\text{gen}}$, and the inferred MEC embeddings $g^{\text{inf}}$ are aligned with the generated MEC embeddings $g^{\text{gen}}$.

- **The transition loss:** The $\mathcal{L}_{\text{transition}}$ constrains the predicted displacement vector $\Delta g_t^{\text{gen}} = f_{\text{forward}}(z_t, g_t^{\text{gen}})$ to be close to the true displacement vector $\Delta g_t^{\text{inf}}$ at time step $t$. The transition loss is computed as the mean squared error (MSE) between the predicted and true displacement vectors. Cosine similarity is also used to ensure the predicted displacement vector aligns with the true displacement vector. To prevent the model from falling into a local minimum during training, where the $g$ alignment loss might make the predicted $g_{t+1}^{\text{gen}}$ closer to $g_t^{\text{inf}}$ rather than $g_{t+1}^{\text{inf}}$, we add an extra contrastive loss term that constrains the distance between predicted $g_{t+1}^{\text{gen}}$ and $g_t^{\text{inf}}$. This is implemented using cosine similarity. The overall form of the loss is:

$$\mathcal{L}_{\text{transition}} = \text{MSE}(\Delta g_{1:T-1}^{\text{gen}}, \Delta g_{1:T-1}^{\text{inf}}) + \alpha \cdot \left(1 - \text{CosSim}(\Delta g_{1:T-1}^{\text{gen}}, \Delta g_{1:T-1}^{\text{inf}})\right) \\ + \beta \cdot \text{CosSim}(g_{1:T}^{\text{gen}}, g_{0:T-1}^{\text{inf}}), \tag{12}$$

where $\alpha$ and $\beta$ are hyperparameters that control the relative importance of the cosine similarity terms.

- **The regularization loss:** To prevent collapse, we utilize the VICReg objectives (Bardes et al., 2022) to regularize $p_t^{\text{inf}}$ and $g_t^{\text{inf}}$. The variance loss encourages the model to maintain a certain level of variance across batches in the latent space, while the covariance loss penalizes the model for having high covariance between different dimensions of the latent space. The overall form of the regularization loss is:

$$\mathcal{L}_{\text{var}}(Z, \gamma) = \frac{1}{TD} \sum_{t=0}^{T} \sum_{j=0}^{D} \max\left(0, \gamma - \sqrt{\text{Var}(Z_{:,t,j}) + \varepsilon}\right) \tag{13}$$

$$\mathcal{L}_{\text{variance}} = \mathcal{L}_{\text{var}}(g^{\text{inf}}, \gamma = 0.5) + \mathcal{L}_{\text{var}}(p^{\text{inf}}, \gamma = 0.5) \tag{14}$$

$$\mathcal{L}_{\text{cov}}(Z) = \frac{1}{D(D-1)} \sum_{i \neq j} (\text{Cov}(Z)_{ij})^2 \tag{15}$$

$$\mathcal{L}_{\text{covariance}} = \mathcal{L}_{\text{cov}}(g^{\text{inf}}) + \mathcal{L}_{\text{cov}}(p^{\text{inf}}) \tag{16}$$

$$\mathcal{L}_{\text{regularization}} = \phi \cdot \mathcal{L}_{\text{variance}} + \psi \cdot \mathcal{L}_{\text{covariance}} + \omega \cdot \mathcal{L}_{\text{smooth}} \tag{17}$$

where $\phi$, $\psi$ and $\omega$ are hyperparameters that control the relative importance of the variance, covariance and smooth(Eq. 8) losses, respectively.

## B.2. Training hyperparameters

We provide the training hyperparameters (Table 7) used in our experiments.

## B.3. Compute requirements

The model is trained on a large dataset (SSv2, 220,000 videos), with training requiring 6-8 hours (10 epochs, parallel training using 3 A100 GPUs). Inference time is very fast due to the relatively small size of the spatial-temporal Transformer and multi-scale VQ-VAE, resulting in minimal overhead. So far, increasing model size has not led to significant time increases.

*Table 7.* Training hyperparameters

| PHASE | PARAMETER | VALUE |
|---|---|---|
| **Fixed parameters** | | |
| | Learning Rate | 1e-4 |
| | Optimizer | AdamW |
| | Weight Decay | 1e-4 |
| | Betas | (0.9, 0.999) |
| | Gradient Clipping | 0.1 |
| | lr_scheduler | CosineAnnealingLR |
| | Epochs | 10 |
| **Stage 1** | | |
| | Batch Size | 32 |
| | Sequence Length | 8 |
| | $\mathcal{L}_{\text{recon}}^{p^{\text{inf}} \to s^{\text{rec}}}$ Weight | 5.0 |
| | $\mathcal{L}_{\text{recon}}^{g^{\text{inf}} \to s^{\text{rec}}}$ Weight | 5.0 |
| | $\mathcal{L}_{\text{alignment}}^{p}$ Weight | 0.22 |
| | $\phi$ (variance loss) | 0.01 |
| | $\psi$ (covariance loss) | 0.01 |
| | $\omega$ (smooth loss) | 0.01 |
| **Stage 2/3** | | |
| | Batch Size | 224/32 |
| | Sequence Length | 2/8 |
| | $\mathcal{L}_{\text{recon}}^{p^{\text{inf}} \to s^{\text{rec}}}$ Weight | 5.0 |
| | $\mathcal{L}_{\text{recon}}^{g^{\text{inf}} \to s^{\text{rec}}}$ Weight | 5.0 |
| | $\mathcal{L}_{\text{gen}}^{g^{\text{gen}} \to s^{\text{gen}}}$ Weight | 3.0 |
| | $\mathcal{L}_{\text{alignment}}^{p}$ Weight | 1.0 |
| | $\mathcal{L}_{\text{alignment}}^{g}$ Weight | 5.0 |
| | $\alpha$ (transition loss) | 1 |
| | $\beta$ (contrastive loss) | 1 |
| | $\phi$ (variance loss) | 0.05 |
| | $\psi$ (covariance loss) | 0.05 |

# C. Dataset details

We aim to learn abstract latent transitions from real-world videos. Unlike 2D games or robot demonstrations, real-world human videos exhibit diverse transitions without explicit action labels. We investigate whether pre-training on large-scale human video datasets enables our model to learn versatile latent transitions that generalize to unseen data. We use the following datasets.

## C.1. Something-Something V2

Something-Something V2(SSv2) (Goyal et al., 2017) contains 220,847 video clips of humans performing actions with everyday objects. We use these large-scale real-world human videos to train our model and maintain the same train/validation/test splits as established in (Goyal et al., 2017).

## C.2. 3D objects primitive transformation datasets

### Rotation datasets
We use three different rotation datasets to evaluate and analyse the model. COIL-100 (Nene et al., 1996) contains images of

100 objects viewed from different angles. MIRO (Kanezaki et al., 2018) is another dataset of 3D object rotations along a different axis.

We also create a synthetic dataset of 3D object rotation containing 5911 objects of 216 daily categories with 72 different views per object. We use Blender to render meshes from the OmniObject3D (Wu et al., 2023), a dataset of high-quality real-scanned meshes, to create 3D rotation objects. Each object mesh is initialized at 0° and then rotated 360° around the vertical axis in 5° increments, yielding 72 rendered views per object. Our dataset covers 216 categories with a long-tailed distribution, incorporating most daily object realms. We include all raw scans provided on the official website, where the number of categories and objects may slightly differ from those reported in the original OmniObject3D paper. The rendering code is adapted from the implementation provided by (Deitke et al., 2023).

**Primitive transformation datasets**
We construct a new dataset using OmniObject3D with sequences containing different primitive transformations (rotation, horizontal/vertical translation, and scaling). Each sequence features an object where the category, initial position, orientation, and size are randomized.

### C.3. Simulated benchmarks

We also evaluate our model on simulated benchmarks to investigate whether the model trained on real-world data can transfer to virtual environments. We use four different simulated datasets: Franka Kitchen (Gupta et al., 2019), Block Pushing (Florence et al., 2022), Push-T (Chi et al., 2023), and LIBERO Goal (Liu et al., 2023).

### C.4. Sequence construction from the 3D objects rotation datasets

Since the model takes image sequences as input rather than single object views, we need to construct sequences using images from the 3D object rotation datasets. We use three 3D object rotation datasets. Here, we describe how to construct sequences from different object views in these datasets.

Each sequence consists of frames of a single object. Between any two adjacent frames, the second frame is a rotated version of the first. The transition here corresponds to the rotation angle: a positive value indicates clockwise rotation, while a negative value indicates counterclockwise rotation. Therefore, we construct a sequence by defining an initial frame and a sequence of rotation transitions; the corresponding images are retrieved from the rotation datasets.

Here are some experiments' sequence construction settings:

- In Section 5.1 Fig. 4(C), the relative rotation transitions are fixed at $5°$, objects in COIL-100 are rotated clockwise around the vertical axis by $5°$ per frame.

- In Section 5.2, we use different fixed parameters: Fig. 5(B), (C), and (F) use rotation angles of $30°$, $20°$, and $30°$ per step respectively; Fig. 5(G) uses fixed scaling of $0.85$ per step.

- In Section 5.3 Table 3, the relative rotation transitions are randomly sampled from $-90°$ to $90°$.

- In Section 5.4 Table 4, the relative rotation transitions are randomly sampled from $-30°$ to $30°$.

# D. Baseline additional results

## D.1. Compute and wall-clock time comparison

*Table 8.* Quantitative comparison of Inference FPS and Average time per batch across different models.

| Model | LAPA | Moto | AdaWorld(LAM) | Our model |
|---|---|---|---|---|
| Inference FPS | 205.33 | 55.22 | 35.60 | 84.00 |
| Average time per batch (s) | 0.623 | 2.318 | 3.595 | 1.523 |

We run a set of experiments on a single NVIDIA A100 GPU to fairly compare the inference throughput. We use a consistent batch size of 16 and a sequence length of 8, averaging the results over 100 batches to calculate the Inference FPS (higher is better) and Average Time per Batch (lower is better).

Our HPC-MEC module adds almost no computational overhead. The reason for this high efficiency is that our module operates entirely in the latent space between the VQ-VAE encoder and decoder. Also, our full model's inference speed is faster than Moto and AdaWorld(LAM). We believe this analysis shows that our model's advantages are achieved without incurring an unreasonable computational penalty.

# E. Ablation study additional results

## E.1. latent transition validity experiment

We conduct another ablation study to examine the role of latent transitions in governing transition dynamics. Recall that the transition dynamics in our model are defined as:

$$\mathbf{g}_{t+1} = \mathbf{g}_t \oplus f_{\text{forward}}(\mathbf{z}_t, \mathbf{g}_t), \tag{18}$$

where $\mathbf{z}_t$ denotes the latent transition, and $f_{\text{forward}}$ integrates content information into $\mathbf{z}_t$ to generate the displacement vector $\Delta \mathbf{g}_t$. Our ablation study focuses on two aspects: the latent transition and content binding. First, we disrupt the input to the inverse model by setting it to zero, and then perform both one-step and autoregressive predictions (Fig. 6(A, B)). We observe that in the one-step prediction, the model collapses to simply copying the previous frame, while in the autoregressive setting, only the first frame is retained, and the sequence shows no meaningful transitions. Second, we impair the content binding by allowing $\mathbf{z}_t$ to be combined only with a zero input to produce $\Delta \mathbf{g}_t$ (Fig. 6(C, D)). In this case, one-step prediction generally preserves the overall transition dynamics, but the generated details are degraded; by comparison, autoregressive prediction yields even poorer results. These findings indicate that the latent transition primarily drives the main transitions, whereas content binding is essential for reconstructing detailed, scene-specific information.

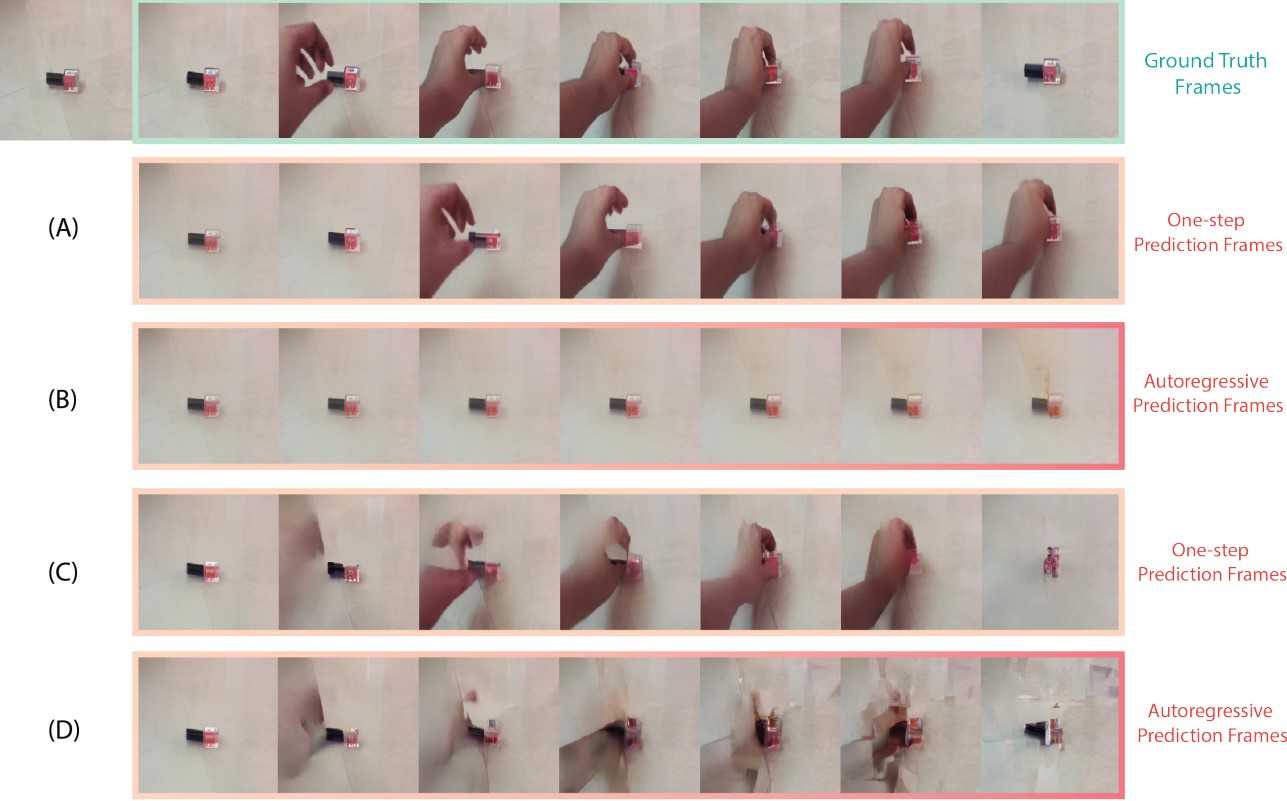

*Figure 6.* **latent transition validity experiment.** (A) The inverse model receives zero inputs, resulting in meaningless latent transitions for one-step prediction. (B) Meaningless latent transition for autoregression. (C) $f_{\text{forward}}$ combines latent transitions with meaningless content information and performs one-step prediction. (D) Meaningless content information binds to latent transitions and performs autoregression.

# F. Additional results

## F.1. Prediction results

### F.1.1. ONE-STEP PREDICTION IN OUT-OF-DISTRIBUTION ROTATION DATASETS

We provide more visualizations of the model's prediction on several rotation datasets. The results are shown in Fig. 7.

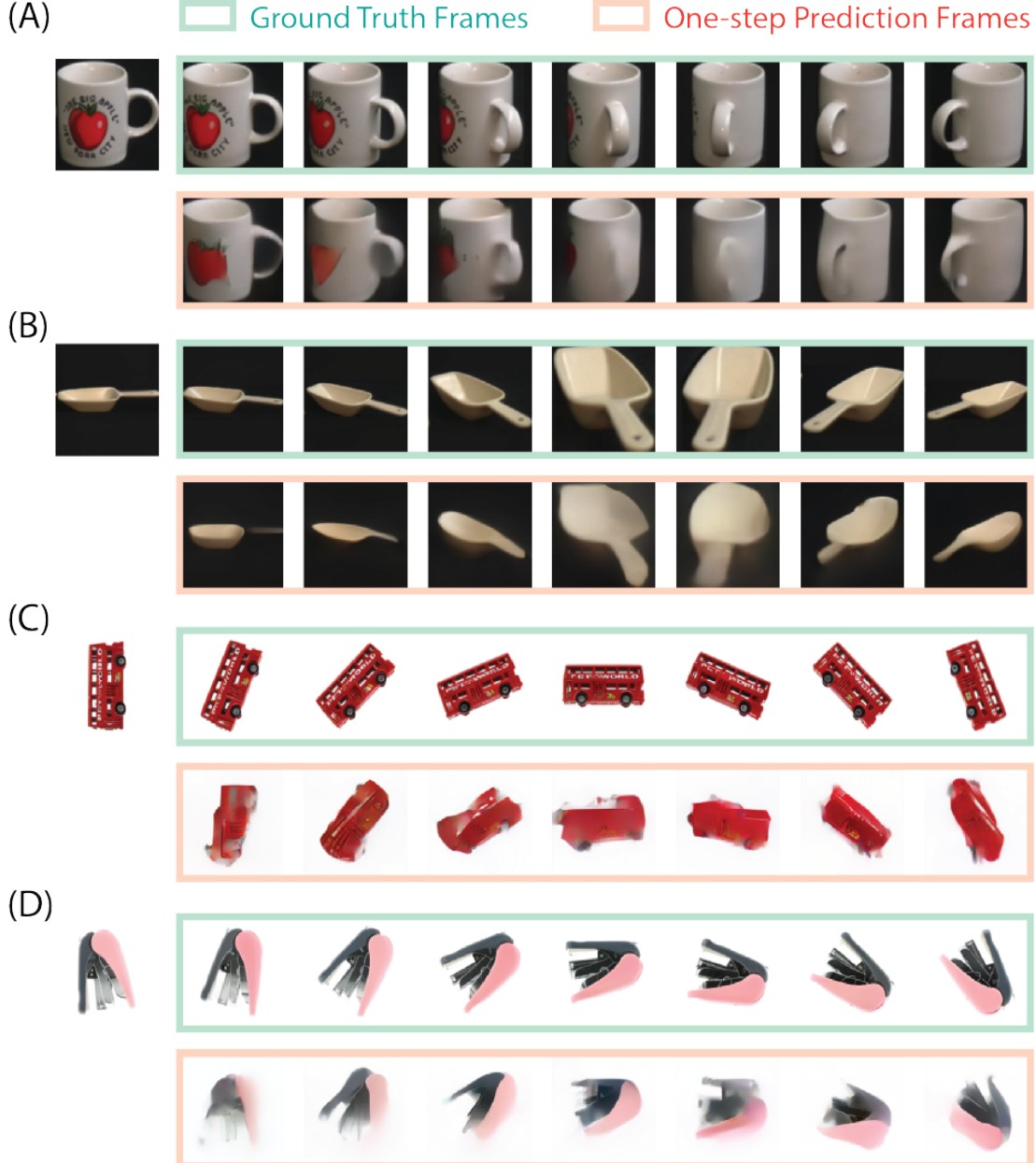

*Figure 7.* **One-step prediction in rotation datasets.** (A, B) One-step prediction evaluated on the COIL-100 dataset. (C, D) One-step prediction evaluated on the MIRO dataset.

F.1.2. ONE-STEP PREDICTION IN OUT-OF-DISTRIBUTION SIMULATED ENVIRONMENTS

We provide one-step prediction results on simulated benchmarks with substantial distributional shifts from human videos in Fig. 8. The model performs robustly in the more naturalistic Franka Kitchen (Gupta et al., 2019), but less effectively in artificial environments like Push-T (Chi et al., 2023) and Block Pushing (Florence et al., 2022).

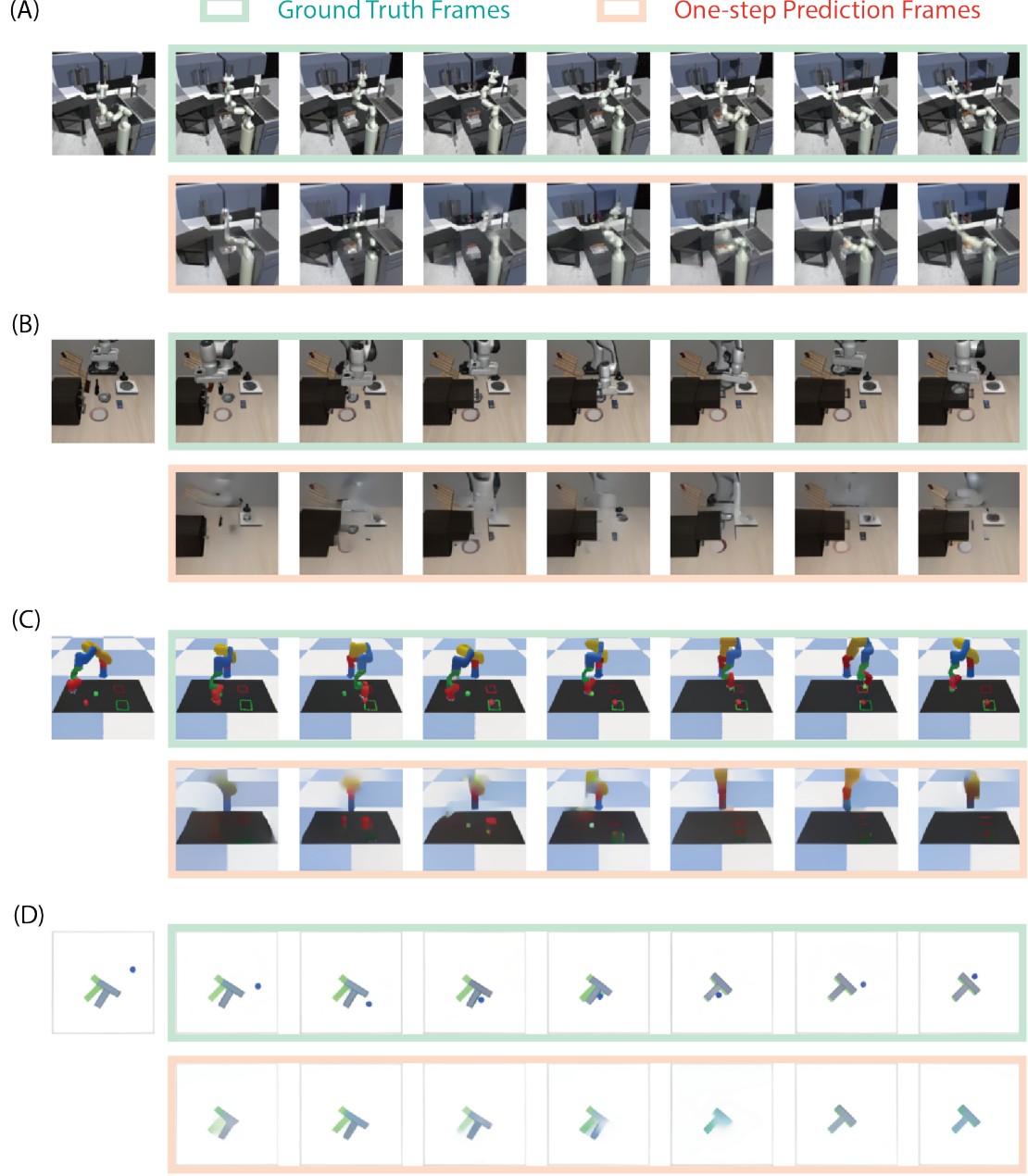

*Figure 8.* **One-step prediction in simulated environments.** (A) One-step prediction evaluated in Franka Kitchen. (B) LIBERO Goal. (C) Block Pushing. (D) Push-T.

## F.2. latent transition reuse results

### F.2.1. ONE-STEP AND AUTOREGRESSIVE REUSE OF LATENT TRANSITIONS ON UNSEEN 3D ROTATION DATASET

We present additional results on OOD 3D rotation dataset to validate robustness in Fig. 9. Here we examine cubic objects, where latent transitions from a source sequence effectively transform frames of a different object, aligning well with the source's rotational dynamics.

**(A)**

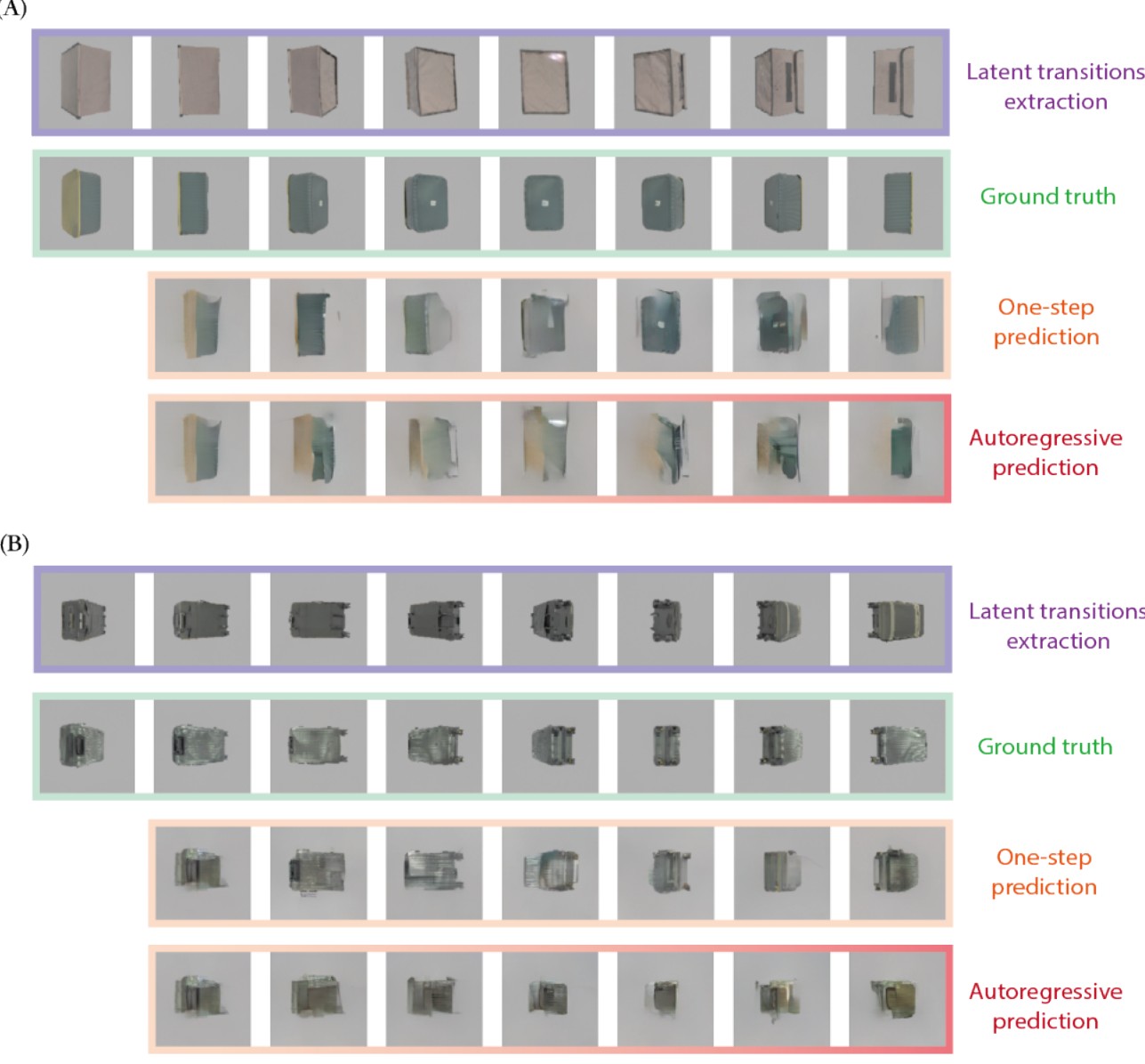

*Figure 9.* **latent transition transfer on OmniRotation.** (A, B) Two examples demonstrating one-step and autoregressive reuse of latent transitions for cubic objects in the OmniRotation dataset.

### F.2.2. ONE-STEP AND AUTOREGRESSIVE REUSE OF LATENT TRANSITIONS ON UNSEEN ARTIFICIAL ENVIRONMENT FRANKA KITCHEN

In Section 4, we demonstrate the model's ability to extract shared latent transitions from sequences of the same action performed under varying contexts in artificial environments. Here, we provide results of one-step and autoregressive reuse of latent transitions in Franka Kitchen (Fig. 10). We observe that the model can effectively transfer latent transitions across different scenes in different contexts, even in such out-of-distribution environments compared to the training dataset.

Additionally, these results suggest that the model can extract content-independent structures from artificial environments, and we believe it has the potential to perform well in generation and reuse tasks with an improved decoder.

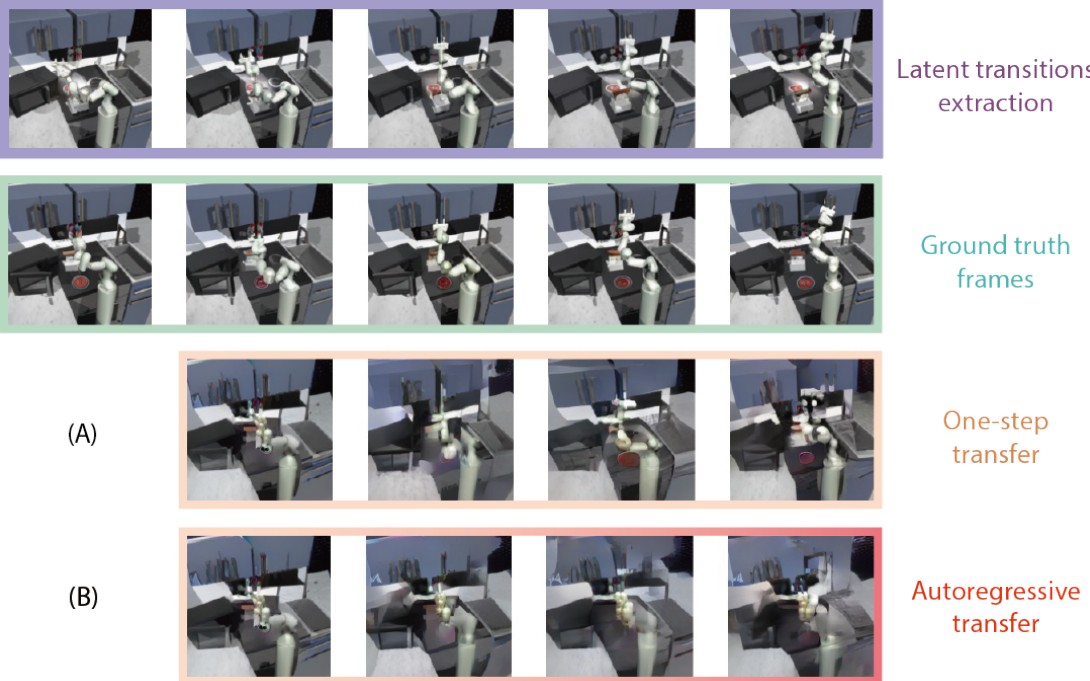

*Figure 10.* **latent transition transfer in artificial environments.** (A) One-step prediction by transferring the sequential latent transitions in Franka Kitchen. (B) Autoregressive prediction by transferring the sequential latent transitions in Franka Kitchen.

### F.3. Visualization of baseline comparison

We provide additional visualization of one-step prediction using LAPA, Moto, AdaWorld(LAM), and our model. The results are shown in Fig. 11. LAPA optimizes directly at the pixel level, resulting in more reliable generation of local details. In contrast, our model is optimized in the latent representation space, enabling it to better preserve overall generation quality even under large action-induced variations. In such scenarios, LAPA's generation quality tends to deteriorate, whereas our model remains robust. Moto and AdaWorld(LAM) all fail to generalize at the Franka Kitchen dataset.

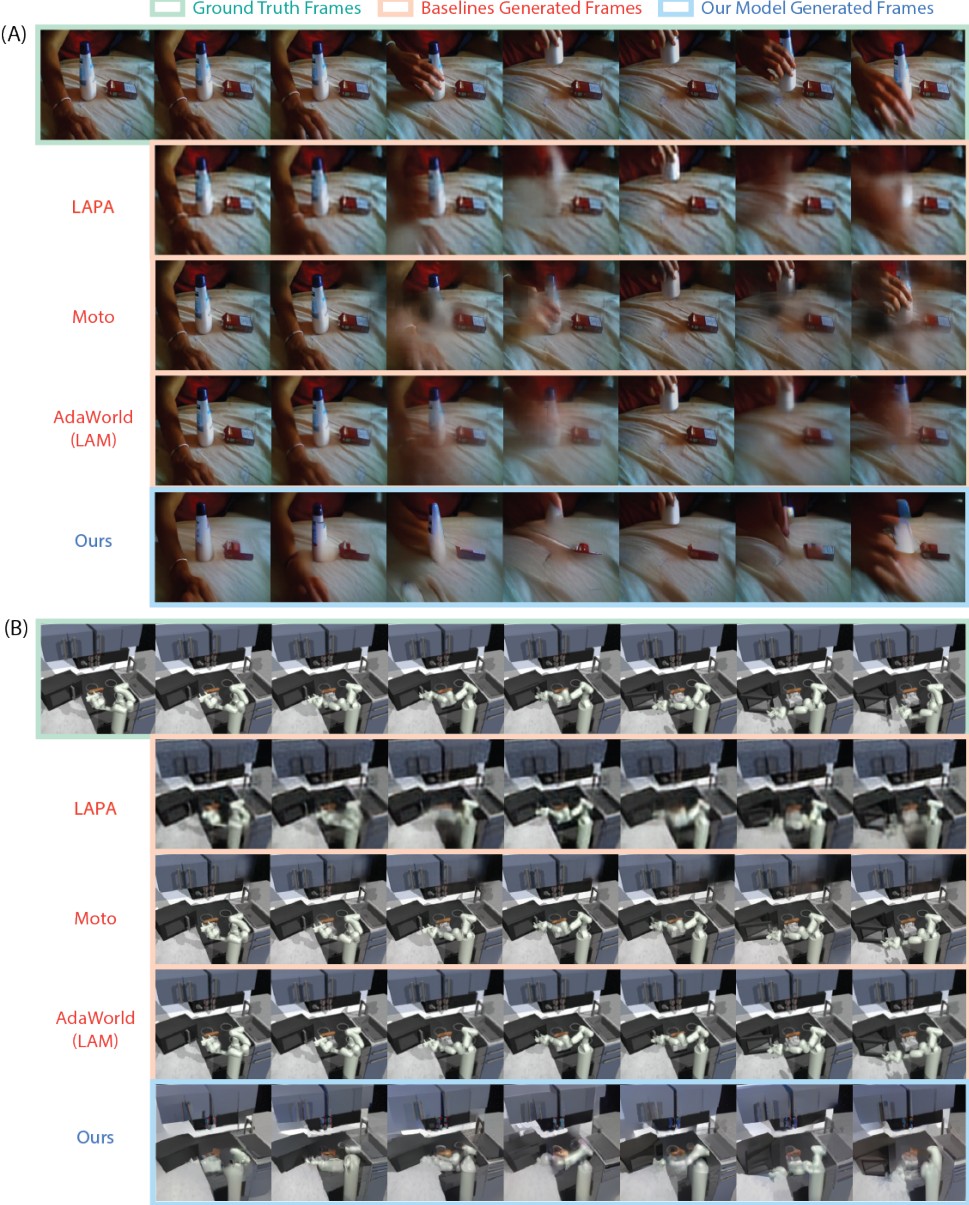

*Figure 11.* **Comparison of generation quality between baselines and our model.** (A) Visualization of one-step prediction on the SSv2 dataset. (B) Visualization of one-step prediction on the Franka Kitchen dataset.

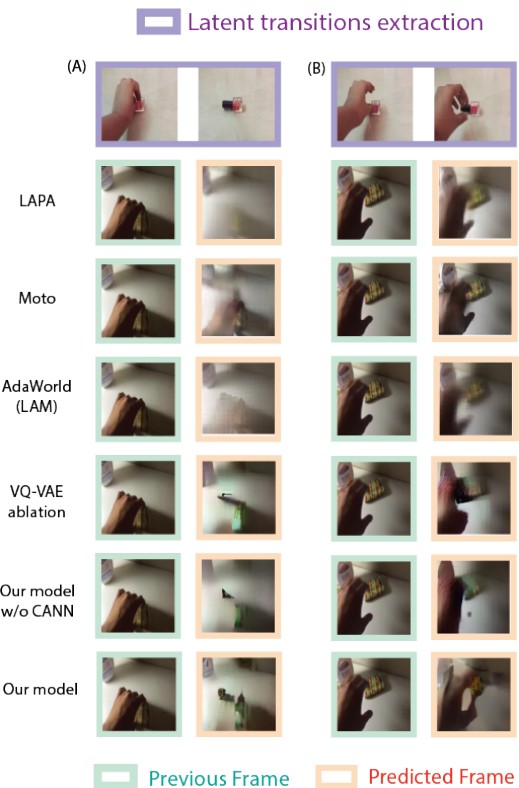

*Figure 12.* **Comparison of latent transition transfer between baselines and our model.** One-step latent transition transfer across different scenes on SSv2, using the same examples as in Fig. 5(A).

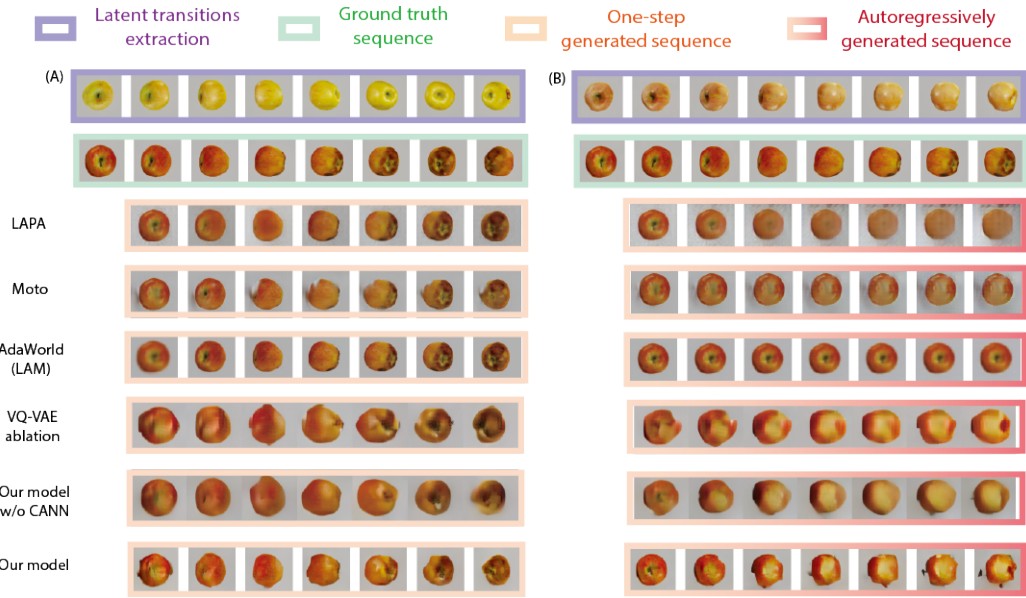

*Figure 13.* **Comparison of latent transition transfer between baselines and our model.** One-step & autoregressive prediction by transferring the sequential latent transitions, using the same examples as in Fig. 5(B, C).

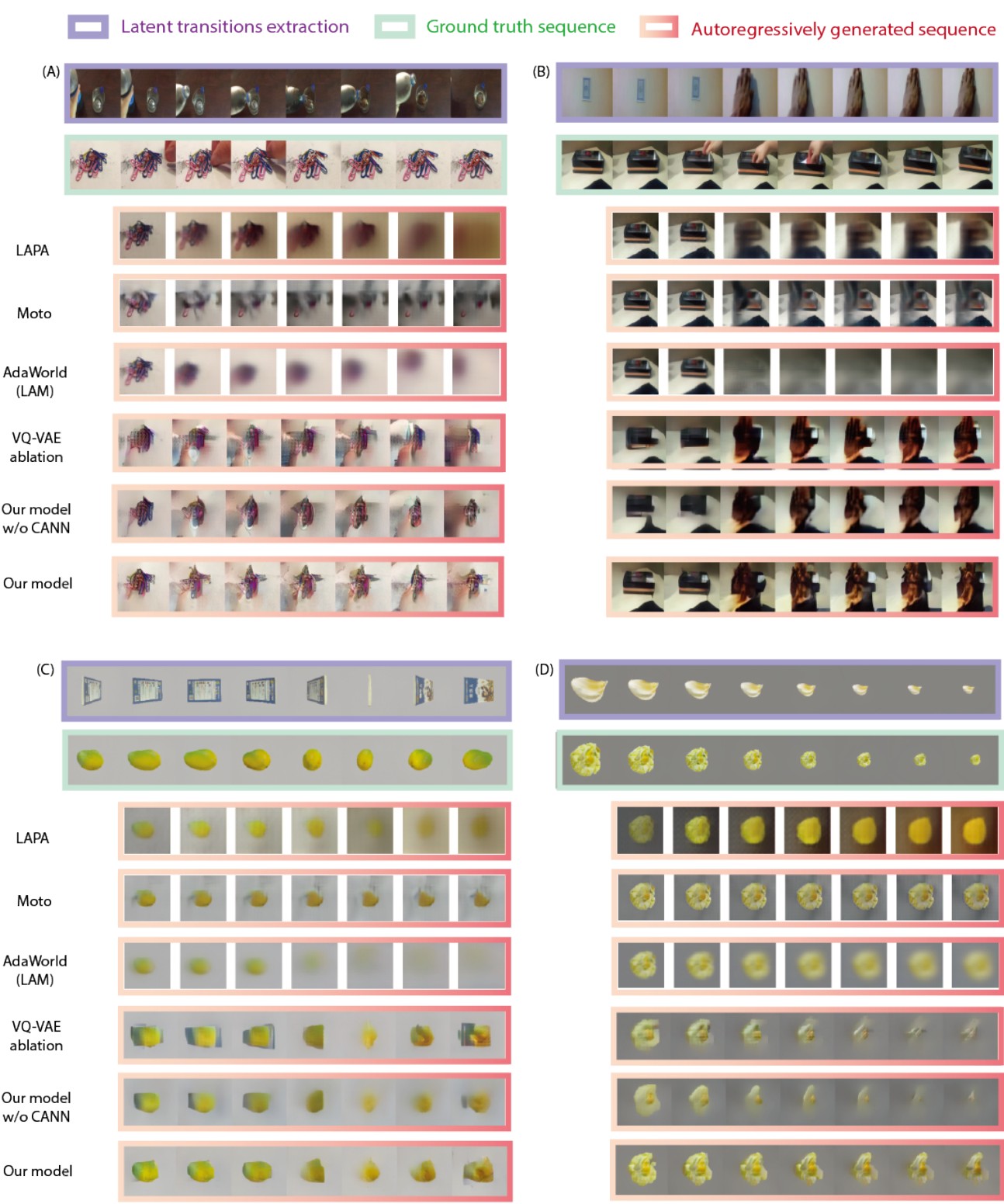

*Figure 14.* **Comparison of latent transition transfer between baselines and our model.** (A)(B)Autoregressive reuse of latent transitions on SSv2, using the same examples as in Fig. 5(D, E). (C)(D)Autoregressive reuse of latent transitions on rotation and scaling dynamics across object categories, using the same examples as in Fig. 5(F, G).

# G. Learned latent space details

### G.1. Dimensionality reduction experiment

In Section 4.1 Fig. 3(A, D), each UMAP visualization shows the embeddings of a single object. In Fig. 3(B), each UMAP figure includes embeddings of 10 pumpkins, 7 red apples, and 3 yellow apples. In Fig. 3(A, B, D), the transition is fixed at $5°$ per step. The objects rotate clockwise around the vertical axis; the sequences in (A) complete two full rotations, while those in (B, D) complete one full rotation each.

### G.2. Category classification decoder

In the decoder in-class structural sharing experiment illustrated in Fig. 3(C), we construct a subset of 500 objects from the 3D object rotation dataset rendered from OmniObject3D (Wu et al., 2023) by randomly selecting 50 categories and then randomly sampling 10 objects from each category. Then we repeat the experiment five times using this subset. In each run, we split the objects in each category into 80% for training and 20% for testing, ensuring that no object appears in both sets. The training and test samples are the per-timestep embeddings extracted from sequences of these training and testing objects.

Each object is used to construct one rotation sequence. Specifically, we initialize the object at a random view and apply a fixed transition of $5°$ per step, meaning the object is rotated clockwise around the vertical axis by $5°$ at each timestep, until a full $360°$ rotation is completed. This results in a 72-frame sequence per object. The sequence is passed through our Hippocampal-Entorhinal-Inspired Coupling Model to extract the **p** and **g** embeddings, which serve as inputs for training the decoder.

The decoder is adapted from the simple latent transition decoder used in (Ye et al., 2024). It is an MLP consisting of two hidden layers with 128 units and ReLU activations. It is trained using the AdamW optimizer with PyTorch's default parameters. We use cross-entropy loss, a batch size of 128, and train for 30 epochs.

Fig. 3(C) reports the average training and test accuracy over the five runs, with the shaded area indicating the standard error.

### G.3. Latent transition decoding details

We construct a new dataset using OmniObject3D with sequences containing different transformations (rotation, horizontal/vertical translation, and scaling). Each sequence features an object where the category, initial position, orientation, and size are randomized. We use this dataset to decode the transformation type from the latent transition.

We use the action decoding paradigm commonly used in latent transition decoding (Schmidt & Jiang, 2024). First, we use the model to extract the latent transition representation from the video sequence, and then feed the latent transition into the decoder (MLP) to decode the transformation type $a_t$, which represents the type of the current transition at time $t$. For each tested model, we train a latent transition decoder following LAPO (Schmidt & Jiang, 2024). Each decoder is implemented as a fully connected network with hidden sizes of 128 and 128.

### G.4. Transition composition analysis

We also provide a transition composition analysis here. We quantitatively test for transition compositionality by decomposing a diagonal movement (move right-down $45°$) into the sum of its constituent horizontal and vertical translations. Using the pumpkins and apples dataset (from Fig. 5(B)), we compare frames generated from the original diagonal latent transition with those generated from the vector sum of the horizontal and vertical latent transition embeddings. Fig. 15 shows that frames driven by compositional latent transitions produce reasonable results comparable to real latent transitions. The results in Table 9 demonstrate strong visual and quantitative similarity:

*Table 9.* Transition composition results comparing real latent transitions with composed latent transitions.

| Transition type | SSIM ↑ | LPIPS ↓ |
|---|---|---|
| Real latent transition | $0.946 \pm 0.021$ | $0.076 \pm 0.032$ |
| Latent transition $A + B$ | $0.944 \pm 0.023$ | $0.073 \pm 0.032$ |

The metrics are statistically comparable (t-test: SSIM p=0.417, LPIPS p=0.402, n=160), providing strong evidence that our latent transition space supports linear composition through its path integration dynamics.

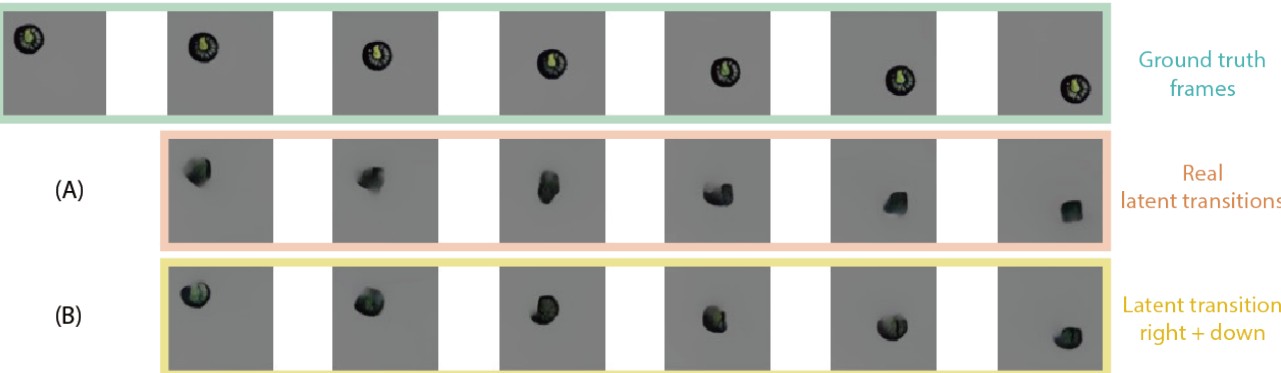

*Figure 15.* **Transition composition results.** (A) One-step prediction frames driven by real latent transitions. (B) One-step prediction frames driven by compositional latent transitions obtained through the summation of rightward and downward latent transitions extracted from the corresponding sequences.

