# OpenReview forum: "Structure Abstraction and Generalization in a Hippocampal-Entorhinal Inspired World Model"
_ICML.cc/2026/Conference — ICML 2026 regular_

### Official Review · Reviewer_ss3H · 2026-03-10

**Soundness:** 3
**Presentation:** 3
**Significance:** 2
**Originality:** 2
**Overall Recommendation:** 4
**Confidence:** 3

**Summary:**

This paper proposes a hierarchical world model inspired by the functional separation mechanism of the hippocampus–medial entorhinal cortex (HPC–MEC) circuit, aiming to simultaneously learn concrete sensory content and generalizable abstract structures from unlabeled high-dimensional visual sequences. The method extracts latent transition variables via an inverse model and introduces dynamics constraints based on continuous attractor neural networks (CANN) to separately model content representations and structural representations. Experimental results show that the model achieves effective disentanglement between appearance and dynamics across multiple datasets and out-of-distribution (OOD) scenarios, and supports cross-object structural transfer.

**Compliance With Llm Reviewing Policy:**

Affirmed.

**Final Justification:**

My concerns have been well addressed, and I am happy to revise my score upward accordingly.

**Key Questions For Authors:**

1. Content–structure disentanglement
The paper claims that HPC captures content information while MEC captures structural dynamics. Could the authors provide more quantitative evidence supporting this disentanglement? (e.g., linear probing on HPC, MEC, and latent transitions to separately predict content and transformation types)

2. Potential dependency on the pretrained visual backbone.
The method relies on a pretrained VQ-VAE encoder and decoder, yet the paper does not analyze how sensitive the performance is to this choice. Additional experiments using different visual encoders (e.g., MAE, DINO features, or randomly initialized encoders) could help clarify whether the improvements mainly come from the proposed hierarchical world model or from strong pretrained visual representations.

3. Limited ablation for the transition dynamics module.
The paper shows that replacing the CANN module with a simple state–action concatenation model degrades performance. However, additional comparisons with stronger transition models (e.g., GRU-based dynamics, Transformer dynamics, or Neural ODE models) would better demonstrate whether the performance gain is specifically due to the CANN structure.

**Limitations:**

yes

**Strengths And Weaknesses:**

Strengths ：
1. Novel biologically-inspired hierarchical world model
The paper proposes a biologically inspired hierarchical architecture that separates content representations (HPC) from structural dynamics (MEC). This design provides an intuitive inductive bias for modeling world dynamics and offers a novel perspective compared to existing latent action or world model approaches that typically entangle content and dynamics.

2. Structured latent transition modeling with CANN dynamics
The introduction of Continuous Attractor Neural Networks (CANN) for modeling latent transition dynamics is technically interesting. The translation-invariant structure of CANN provides a meaningful inductive bias for modeling continuous state transitions and path integration, which aligns well with the goal of capturing abstract motion dynamics.

3. Empirical evidence of structural generalization
The experiments demonstrate that the learned latent transitions can be reused across different objects and contexts, suggesting that the model captures transferable structural dynamics rather than merely memorizing appearance patterns. This property is particularly relevant for generalization in world models and embodied AI.

Weaknesses:
1. Limited quantitative evaluation of representation properties.
While the paper argues that the model separates structural dynamics from visual content, most supporting evidence is based on visualization and qualitative analysis. More quantitative evaluation of representation properties would strengthen the claims.

2. Some experimental choices are insufficiently justified.
Certain design choices in the model architecture and training pipeline (e.g., the specific dynamics module design and reliance on pretrained visual encoders) are not extensively analyzed in the experiments, making it harder to fully understand the source of the performance gains.

---

> ### Author Rebuttal · Authors · 2026-03-30
>
> We sincerely thank the reviewer for the positive assessment of the paper’s novelty, the biologically grounded hierarchical design, and the evidence for structural generalization. We address the main concerns below.
>
> W1/Q1: We agree this is an important point. As demonstrated in our experiments, we provide several quantitative analyses beyond visualization:
> 1. Fig. 3(C) provides **a quantitative probe of HPC vs. MEC embeddings**: classifiers trained on these embeddings show that MEC representations generalize better across held-out objects, while HPC embeddings retain more instance-specific variation. This is consistent with our claim.
> 2. Table 3 includes **a transition-decoding benchmark**, where an MLP predicts transformation type from the learned latent transition. Our model achieves the best accuracy among all baselines and ablations, providing direct evidence that its latent transitions are better aligned with transformation types.
> 3. Appendix G strengthens this further: **transition decoding accuracy** is highest from $z$, then $\Delta g$, and much lower from $p$ or raw $g$, showing that transformation information is indeed concentrated in the learned latent transition rather than entangled uniformly across representations.
> 4. We also contain **a quantitative reuse metric $R$ for latent transition transfer**. The unified-latent-space baseline shows lower $R$ than our hierarchical model, and we explicitly interpret the unified baseline’s “texture leakage” as evidence that entangling content and transition structure harms transfer. Our full model quantitatively supports better content/structure separation for reuse.
>
> We agree that the manuscript should make this evidence more explicit, and in the revision, we will reorganize these results to present them more directly as evidence of disentanglement. We would also like to clarify that probing “content” is less straightforward than probing transformation type, because “content” does not have a clean, well-defined label in our setting. For this reason, we use the quantitative comparison in Fig. 3 to distinguish their representational roles.
>
> W2: Thank you for this valuable question. As demonstrated by our controlled ablations, the performance gains can be explicitly decomposed into:
> 1. **Hierarchical factorization matters**: the unified latent-space ablation shows worse reuse and more content leakage (Table 2).
> 2. **CANN dynamics matter**: removing CANN reduces reuse quality, OOD transfer, and transition semantics (Table 2 & 3).
> 3. **The gains are not from VQ-VAE alone**: the VQ-VAE ablation underperforms substantially despite using the same pretrained visual front-end (Table 2 & 3 & 4).
> 4. **The learned transitions are semantically cleaner**: the transition decoder gives the highest accuracy for our full model (Table 3), and Appendix G shows that the strongest transition information is concentrated in $z$.
>
> Q2: We appreciate this insightful suggestion. While evaluating different encoders (e.g., MAE/DINO) is a valuable future direction to test the framework's encoder-agnosticism, our **controlled VQ-VAE ablation** directly answers your core concern: **whether the performance gains stem from the proposed hierarchy or simply the strong pretrained backbone.**
>
> By fixing the identical pretrained VQ-VAE and omitting only the HPC-MEC hierarchy (using a parameter-comparable Inverse + forward model), we strictly isolate the structural contribution. Compared to this ablation, our full model significantly improves:
> - Latent transition reuse ratio $R$: 2.035 $\to$ 3.201 (1-step), 1.796 $\to$ 2.482 (autoregressive).
> - LPIPS: 0.158 $\to$ 0.120 (1-step), 0.177 $\to$ 0.156 (autoregressive).
> - Transition decoding accuracy: 0.8523 $\to$ 0.9064.
>
> Crucially, the VQ-VAE ablation completely fails the OOD transfer task. If our performance relied solely on the visual backbone, transition reuse would not collapse when the hierarchical dynamics are removed. We will make this isolation logic explicit in the revision. Therefore, the pretrained backbone is helpful for stable visual representations, but it does not explain the observed gains; these gains arise from the proposed hierarchical structural world model.
>
> Q3: We appreciate this suggestion. We would like to clarify that our claim is **not** that CANN is a better temporal model than RNNs or Transformers, but that **its path-integration inductive bias** is beneficial for modeling **state–action interaction** and structural reuse. Since temporal dependency in our model is already handled by the causal Transformer, the relevant comparison is therefore with a standard state–action concatenation module, which is exactly what our w/o CANN ablation tests.
>
> Our ablation explicitly supports this claim. “w/o CANN” ablation consistently worsens reuse quality, LPIPS, and transition decoding accuracy, and it also fails on the OOD latent transition transfer task. This is crucial because it shows that CANN improves structural transfer.

---

> > ### Author Rebuttal · Reviewer_ss3H · 2026-04-02
> >
> > Thank you for your detailed response. My concerns have been well addressed, and I am happy to revise my score upward accordingly.

---

> > > ### Author Response · Authors · 2026-04-03
> > >
> > > It is deeply encouraging to know our response has addressed your concerns. We sincerely thank you for revising the score upward and for your constructive feedback, which has been instrumental in improving this work. We remain committed to further strengthening the paper wherever possible.

---

### Official Review · Reviewer_CY9q · 2026-03-10

**Soundness:** 3
**Presentation:** 3
**Significance:** 2
**Originality:** 2
**Overall Recommendation:** 3
**Confidence:** 3

**Summary:**

The paper addresses how neural systems can learn world models that decouple content and structure to obtain better generalization in video sequences.  The work is inspired by brain systems such as HPC and MEC and consists of multiple components that are trained serially (a pretrained VQ-VAE, followed by two phases of training additional components). The paper shows that best performance from the model is obtained with all proposed components via ablation experiments, and comparisons to other world modeling architectures indicate strong performance from the proposed model.

**Compliance With Llm Reviewing Policy:**

Affirmed.

**Final Justification:**

The article foregrounds the neuroscientific literature to "inspire" a novel architecture, but I remain unconvinced that the inspiration goes much beyond a labeling of layers in a transformer. In the discussion period, the authors did bring up facts about the relative size/complexity of MEC and HPC. It therefore seems essential to argue that (1) in the brain, MEC is a more compact representation than HPC, and (2) in your model, the reduction of dimensionality is critical to the model's success.  But even then, I really would hope for stronger inductive bias from the brain to inform the design of a model before pitching it as neuroscience inspired.

**Key Questions For Authors:**

I am having a difficult time appreciating the contribution of this work beyond the claim of better performance on several tasks than 3 existing world-modeling methods. Are there insights for neuroscience? Are there claims about how the neuroscience has informed model architecture?  Other than the neuro labels given to layers of the transformer, the model looks like a fairly standard amalgamation of prediction and inference components.

There is an elaborate story about the HPC and MEC and their distinct roles, yet this story doesn't seem to go beyond using this jargon to label distinct blocks of a transformer.  What insights am I supposed to gain about brain structures, or in what manner do brain structures inform the modeling being done here? What is the contribution here beyond showing that deeper layers of a transfomer encode slightly more structural information (as one would expect)? (Table 1 does not give any indication of a qualitative difference, which I was hoping for given the introduction.)


The attractor net seems to have been designed specifically to model continuously moving objects in space. Am I correct? How much does the inductive bias in these decisions influence the model's performace?  Could the sort of inductive bias here be incorporated into other models?  Is it a novel contribution of the work?


I'm not sufficiently expert to know that the three methods (LAPA, Moto, AdaWorld) are indeed state of the art. I defer to more savvy reviewers. However, given the hype that JEPA has received in recent years, I would have expected it to be a contender.

How does the model's correction of visual state relate to (a) inference in a Kalman filter, and (b) the type of corrector mechanism proposed in models of learning from video (e.g., Elsayed et al., 2022 https://arxiv.org/abs/2206.07764)

**Limitations:**

Yes

**Strengths And Weaknesses:**

The manuscript is well written and the model and training procedure seems well described.

The strength I see in the work is the empirical results, although (see questions below) I'm not sufficiently savvy about the field to be certain the data sets and alternative models represent state of the art.

The paper emphasizes the unique roles of HPC and MEC in terms of the content-structure distinction. I was expecting the authors to propose some type of inductive bias that yielded differences in function, yet these two structures are identical and implemented as causal transformer blocks.

The weakness I see is a complex architecture and 5-term loss function trained in multiple stages, and unfortunately, I'm not able to glean generalizable insights.  While the particular architecture and loss may be unique, it does seem similar in spirit and functional components to other unsupervised video prediction models.

---

> ### Author Rebuttal · Authors · 2026-03-29
>
> We thank the reviewer for the careful reading and for clearly articulating the central concern. Below, we address the main concerns in detail.
>
> **HPC/MEC distinction and neuroscience contribution** (W1/Q1/Q2)
>
> Our claim is not that the model is a biological implementation of HPC and MEC. Rather, the neuroscience contribution is **computational and normative**. Substantial experimental evidence supports a functional division: HPC binds content-specific information from experiences, while MEC encodes abstract structures. This separation enables structural generalization, allowing the system to bind extracted structures flexibly with novel contexts. Existing cognitive-map models, such as TEM and VectorHaSH, already motivate a similar functional hierarchy, but they cannot infer shared abstract structure from real-world video. **Our work addresses that gap**. Thus, the neuroscience-facing insight is not “this is how the real brain is implemented,” but rather that:
> - The HPC/MEC division can be made computationally concrete in a modern world model.
> - Functional hierarchy avoids the trade-off between pixel-level reconstruction and abstract structure reuse.
> - Velocity-like latent transitions plus grid-cell-like path integration provide a plausible route to structural reuse.
>
> We view this as a meaningful bridge: many neuroscience-inspired models stop at analogy, while many modern world models stop at performance. Our contribution is to show that **the HPC–MEC functional decomposition is a useful inductive principle for learning reusable transition structure from real videos**.
>
> Although both HPC and MEC are implemented using causal Transformers, their roles are not identical. They differ in three key ways. First, they operate at different abstraction levels and dimensionalities: HPC is larger, while MEC is more compressed. Second, only MEC participates in **inverse-model transition learning and path integration**. This latent transition autoencoding process forces the MEC embeddings to be more abstract and smoother. Third, the training objectives explicitly push them toward different functions.
>
> **What generalizable insight do we provide?** (W2/Q2)
>
> The generalizable insight is not the exact recipe of five losses and three stages; it is the principle of organizing video world modeling around a content/structure hierarchy with grid cell-inspired latent transition dynamics. The most generalizable takeaway is that **transition structure is easier to reuse when it is extracted from a grid cell-informed compressed latent space rather than from a unified latent space entangled with content**. This is exactly what the unified-space ablation tests and what the transfer experiments in Fig. 5 / Table 2 support.
>
> We agree that Table 1 alone should not be overclaimed. Its role is to show a **relative increase in abstraction** from $p$ to $g$ to $z$ in a difficult OOD robotic setting. The stronger evidence comes from Fig. 3 and the decoding analyses, where HPC preserves more content-specific variation, and $z$ is the most semantically concentrated transition representation.
>
> Q3: We would like to clarify that a continuous attractor does not equate to simply "moving objects." Rather, it provides **a structured, continuous latent geometry** to organize smooth transitions. Neuroscience evidence shows grid cells form this attractor manifold, enabling path integration in abstract spaces and facilitating mental simulation. This inductive bias is valuable because even complex real-world dynamics can be effectively modeled as displacements on a latent manifold. Consequently, the primary benefit of our CANN-inspired design is enabling zero-shot transition reuse. The "w/o CANN" ablation completely fails OOD transfer, whereas the full model succeeds. We view this structured latent geometry as a **core novelty** that could readily benefit other latent world models, and we are the first work to use CANN-informed dynamics to extract real-world abstract transitions.
>
> Q4: We selected LAPA, Moto, and AdaWorld as baselines because they align with our focus: extracting and reusing latent transitions from observation-only video. V-JEPA learns via masked joint-embedding; it does not infer reusable latent transitions, making it less comparable to our setting. We will clarify this distinction in the revision.
>
> Q5: The visual feedback is a biologically grounded error-correction mechanism. Like a Kalman filter, our model functionally works as a prediction-and-correction process. However, it is not a Kalman filter in the probabilistic sense. We are not maintaining an explicit Gaussian belief state, covariance update, or optimal Bayesian estimator.
>
> It is also different from Elsayed et al. Their “corrector” updates object slots from the current frame, whereas our visual feedback corrects accumulated drift in path integrations. So the relation is loose: both use observation-based correction, but the underlying target and motivation are different.

---

> > ### Author Rebuttal · Reviewer_CY9q · 2026-04-03
> >
> > I thank the authors for their detailed response. I am more confused than ever, which might be due to the time that has passed since I looked at the paper in detail. However, the structures you've labeled HPC and MEC are just shallower and deeper transformer layers. The reason why I'm obsessing over inductive bias that comes from the neuroscience is that you can take any model with shallow and deep layers and give them separate labels.  The _functionally_ distinct components of the model seem to be the HPC+MEC versus the forward model.
> >
> > If I'm wrong, please correct my stupidity.

---

> > > ### Author Response · Authors · 2026-04-05
> > >
> > > We sincerely thank the reviewer for the follow-up, and we really appreciate you sharing your thought process so openly.  Since they share the same backbone type, HPC and MEC can understandably look like “shallower vs. deeper transformer layers.” But that is not the distinction we intend to draw, and we will explain the difference between HPC and MEC in detail. In short, the distinction in our model is not “shallow vs. deep,” but **content-preserving state vs. transition-structured state**.
> > >
> > > We understand the intuition behind the comment: deeper Transformer layers often capture higher-level features. However, our target is not generic semantic abstraction, but **transition abstraction and generalization**. The latent transition must be **content-invariant yet still reusable** for prediction, and this does not emerge trivially from an end-to-end video-prediction Transformer, even if it is very deep. In a unified latent space, the model must both reconstruct full frames and encode abstract transitions. These two goals create a real **trade-off**: content-rich reconstruction and abstract dynamics push the representation in different directions. Our motivation for separating HPC and MEC is exactly to resolve this tension: one pathway preserves richer content for reconstruction, while the other learns an abstract scaffold from which latent transitions are inferred. **This is why our result is not equivalent to simply taking a deeper decoder-only Transformer and naming earlier and later layers differently.**
> > >
> > > The key point is therefore that HPC and MEC are not defined by depth, but by **function, training role, and dimensionality.**
> > > 1. **Function**:
> > > The most important distinction is that **only MEC participates in the transition-learning loop**. As shown in Fig. 2, the inverse model infers latent transitions from **successive MEC embeddings**, and the forward model updates the MEC state using those latent transitions to predict the next latent state. In that sense, the MEC pathway is the locus of **path-integration-style dynamics**. While the HPC pathway does not play that role, it serves as the richer, history-dependent content pathway that supports reconstruction.
> > > Even if we removed the neuroscience labels entirely, the architectural fact would remain: there is a content pathway and a structure/transition pathway, and only the latter is used to infer and apply reusable latent dynamics.
> > > We use Transformer backbones in both places because both HPC and MEC involve temporal processing in the neuroscience literature; however, in our model, they are placed in different computational roles, not simply different depths of the same stack.
> > >
> > > 2. **Training role**: The separation is also enforced by the alignment losses.
> > > The alignment between $\text{p}\^{\text{inf}}$ and $\text{p}\^{\text{gen}}$ requires the HPC latent space to remain recoverable from generated MEC states while preserving content-rich detail.
> > > The alignment between $\text{g}\^{\text{inf}}$ and $\text{g}\^{\text{gen}}$ requires the MEC latent space to focus on dynamics-relevant features for prediction under an information bottleneck.
> > >
> > > 3. **Dimensionality**: HPC first maps visual embeddings into a larger, content-rich state, while MEC compresses this into a lower-dimensional, structure-oriented state. In the appendix, HPC has a hidden size of 8192 (per-patch 512), whereas MEC has a hidden size of 4096 (per-patch 256). So MEC is not simply a “deeper layer”; it is a separate bottlenecked state space with a different role.
> > >
> > > Each major component corresponds to a specific hypothesis, tested via targeted ablations. Crucially, our trained ablations (1 & 2) retain “shallower and deeper layers”, isolating our functional gains:
> > > 1. **HPC–MEC separation.**
> > > We compare against **a unified latent space baseline**, which is **close to the alternative the reviewer is suggesting**. That variant shows worse latent-transition reuse and stronger texture leakage, while the full model achieves the best similarity ratio, SSIM, and LPIPS in Table 2. This supports the need to separate a content-rich pathway from a structure-oriented one.
> > > 2. **CANN-based MEC dynamics.**
> > > We replace the CANN-inspired mechanism with a standard state-action concatenation module (“w/o CANN”). This ablation underperforms the full model and fails on OOD latent-transition transfer, showing that the path-integration inductive bias is useful beyond generic temporal modeling.
> > > 3. **Inverse latent-transition learning.**
> > > In Appendix E, we show that latent transitions drive the main dynamics, while content binding refines scene-specific reconstruction.
> > >
> > > So our intended claim is not that HPC and MEC are special simply because they use different Transformer blocks. Rather, the claim is that **a functional separation between content and structure**, combined with explicit transition learning on the structure pathway, is a useful inductive bias for learning reusable latent dynamics from raw videos.

---

### Official Review · Reviewer_z65m · 2026-03-11

**Soundness:** 2
**Presentation:** 3
**Significance:** 2
**Originality:** 3
**Overall Recommendation:** 4
**Confidence:** 2

**Summary:**

This paper proposes an HPC-MEC inspired hierarchical world model that separates content (HPC) from abstract structure (MEC) using a CANN-based path integration mechanism and an inverse model for latent transition inference. Trained self-supervised on SSv2 video, it is evaluated on 3D rotation datasets and simulated robotic environments for structural abstraction and zero-shot generalization.

**Compliance With Llm Reviewing Policy:**

Affirmed.

**Final Justification:**

Thanks for the thorough rebuttal, which addresses most of my concerns, especially in the fairness of comparison and missing baselines. Hence, I raise my score from 3 to 4

**Key Questions For Authors:**

1. Given the small CANN ablation gap (0.902 --> 0.894 SSIM), what is CANN's actual core advantage over a standard residual MLP?
2. When source and target action types differ substantially, e.g., "push" transitions applied to "rotation", how does structural transfer perform?
3. How frequent must visual feedback be to maintain acceptable autoregressive quality? What does this imply for planning utility?
4. Can you compare your method with more representative baselines?

**Limitations:**

yes

**Strengths And Weaknesses:**

### Strengths

* Bridging cognitive map theory (HPC-MEC circuit) with video world models is novel. The idea of using CANN attractor dynamics for path integration and treating latent transitions as velocity operators is conceptually appealing.
* Structural abstraction analysis (UMAP, periodicity), one-step and autoregressive prediction, cross-object structural generalization, OOD transfer, and comparisons with LAPA/Moto/AdaWorld. Ablations cover CANN removal, unified latent space, and VQ-VAE variants.

### Weaknesses

* The core formula $g_{t+1} = g_t + MLP(z_t, g_t)$ is a standard residual update. Calling this "CANN-inspired" overstates the biological correspondence, it is closer to a Transformer with smoothness regularization. The ablation in Table 2 shows that removing CANN only drops SSIM from 0.902 to 0.894; the limited marginal value suggests that this method may not really resolve the problems.
* Only tested on simple geometric transformations (rotation, translation, scaling), experiments in real-world or complex cases might be required. Whether the model captures semantic-level structure (causality, temporal dependencies) is unverified. The claim is closer to "transformation-type disentanglement."
* *The model uses a pretrained VQ-VAE (VAR, depth=16) frozen during training, while baselines use different visual encoders. Performance gaps in Table 4 may largely reflect encoder quality differences, not the HPC-MEC module.
* Fig. 4 (B) shows quality collapses without visual feedback (GT observations). We might need real observations every few steps, which fundamentally limits the model's utility as a world model for planning in real-world cases.
* **Missing baselines*: No comparison with representative world models like Dreamer(V3) [1], JEPA [2].

[1] Hafner, Danijar, et al. "Mastering diverse domains through world models." arXiv preprint arXiv:2301.04104 (2023).

[2] Assran, Mahmoud, et al. "Self-supervised learning from images with a joint-embedding predictive architecture." Proceedings of the IEEE/CVF conference on computer vision and pattern recognition. 2023.

---

> ### Author Rebuttal · Authors · 2026-03-29
>
> We sincerely thank the reviewer for the thoughtful and constructive assessment. Below, we address the main concerns in detail.
>
> W1/Q1: Thank you for this crucial question. This residual form arises because any first-order continuous dynamical system becomes a residual-style update after Euler discretization. After Euler discretization, a standard CANN is mathematically a residual update (Appendix A.4). The key point is that our $f_{\text{forward}}(z_t, g_t)$ is **not** an arbitrary deep-learning nonlinearity or a hand-designed recurrent kernel. Functionally, it approximates the joint effect of the recurrent dynamics and transition input by decoding a **displacement vector** from the current MEC state and the latent transition, i.e., a path integration task. In our framework, the latent transition is learned through **an inverse-model autoencoding process**: the model first extracts $z_t$ from the change $g_{t+1}-g_t$, and then combines $z_t$ with $g_t$ to reconstruct the corresponding displacement. This is the central difference from a generic “Transformer + temporal smoothness” formulation: the model implicitly learns a **transition autoencoder using a path integration task**, rather than only smoothing latent trajectories. That is our central methodological point.
>
> Furthermore, SSIM is a local image metric that does not fully capture our primary objective: **the structural transfer and reuse of latent transitions**. The CANN-inspired MEC is designed to establish a structured latent space for OOD transfer and stable path integration, rather than merely improving reconstruction fidelity. Therefore, the SSIM results in Table 2 must be interpreted alongside our transfer capabilities. Crucially, both the "w/o CANN" and VQ-VAE ablations completely failed the OOD transfer task. Table 3 demonstrates that the full model achieves the highest action decoding accuracy, outperforming the “w/o CANN” ablation. Therefore, the inductive bias of CANN achieves a much larger gain on the **reusable and transferable abstract dynamics.**
>
> W2:  Thank you for this important question. We would like to clarify that the model is **trained only on SSv2**, yet it already learns to **zero-shot extract** both simple transformations and more complex transitions in OOD robotic environments, reflecting true semantic-level structure. We intentionally use clean transformations such as rotation for analysis because they make the structure/content separation visually interpretable. Our model handles SSv2 interactions—such as object removal (Fig. 5A) and partial occlusions (Fig. 5D/E)—better than baselines (Figs. 12, 14), while also extracting **OOD robotic dynamics** in Franka Kitchen (Table 1). Furthermore, rather than simple 2D pixel shifts, our 3D rotations autoregressively generate physically plausible illumination changes (Fig. 5C). These combined capabilities demonstrate that our model significantly exceeds mere geometric transformations.
>
> W3: This is a good point, and we address your fairness concern through a controlled **VQ-VAE ablation** that isolates the contribution of our HPC-MEC module. This variant uses the identical pretrained VQ-VAE backbone, an inverse model, and an MLP-based forward model of comparable size, but omits the hierarchical HPC-MEC. As Tables 2 and 4 demonstrate, the full model consistently outperforms this ablation, proving our hierarchical mechanism successfully navigates the **trade-off between pixel-level generation and latent abstraction**. More importantly, the VQ-VAE ablation completely fails the OOD transfer task. If our performance relied solely on the visual backbone, we would not observe such a failure in transition reuse.
>
> W4/Q3: The visual feedback is **not a workaround that undermines planning; rather, it is a biologically grounded error-correction mechanism**. This aligns with cognitive-map models like TEM and VectorHaSH, which similarly stabilize internal state estimates using sensory observations. Consequently, our model is highly suited for closed-loop model-based agents interacting with environments that use collected observations to correct accumulated errors in the dynamics model. While studying the exact trade-off between correction frequency and planning efficiency is an important future direction, it remains beyond the scope of this paper.
>
> W5: We selected LAPA, Moto, and AdaWorld as baselines as they align with extracting and reusing latent transitions from observation-only video. DreamerV3 requires ground-truth actions for forward dynamics. V-JEPA learns representations via masked joint-embedding; it does not infer reusable latent transitions. We will clarify this distinction in the revision.
>
> Q2: Our method transfers source dynamics onto target content regardless of the target's original dynamics. Figures 5A/D/E demonstrate this successful transfer. Furthermore, Appendix Figures 12 and 14 show that our model faithfully executes this cross-content transfer, whereas the baselines fail.

---

> > ### Author Rebuttal · Reviewer_z65m · 2026-04-03
> >
> > Address most of my concerns; Will raise my score

---

> > > ### Author Response · Authors · 2026-04-03
> > >
> > > Thank you for your encouraging follow-up. We are grateful that our responses were able to resolve most of your concerns, and we sincerely appreciate your time, feedback, and consideration in reevaluating our work.

---

### Official Review · Reviewer_h2Ak · 2026-03-14

**Soundness:** 3
**Presentation:** 4
**Significance:** 3
**Originality:** 3
**Overall Recommendation:** 5
**Confidence:** 3

**Summary:**

This paper proposes a hierarchical, model inspired by the hippocampal-entorhinal (HPC-MEC) circuit in the brain. Its goal is to infer latent transitions while simultaneously building a visual world model by extracting abstract structures from sequence data in a self-supervised way. The model combines a visual encoder (VQ-VAE), an inverse model for learning latent transitions, and a hierarchical world model (HPC and MEC embeddings) that learns structures from sequences while learning to predict the next frame through velocity-driven path integration.

Experiments using video datasets and simulations show that the model is able to decouple appearance from dynamics and transfer motion structure between objects or different environments without any prior constraints on the underlying dynamics. The authors claim that the model is capable of robust flexibility and OOD generalization thanks to the integration of the encoding done in the MEC with content details stored in the HPC.

**Compliance With Llm Reviewing Policy:**

Affirmed.

**Final Justification:**

I am happy with the authors' responses and am adjusting my score to a 5.

**Key Questions For Authors:**

Could the authors clarify what exactly is meant by "abstract structures" earlier on in the text? (e.g., l. 56) I only understood it much later in Section 4. Providing a concrete example at the beginning of the paper would help.

Please help me understand Fig. 3A: Why does the object with 3 white sides and 1 brown side have a period of 1/4? And why should its low-dim embedding form "two overlapping small circular trajectories, with the remaining half period forming a larger circular trajectory"? This result didn't make sense to me: if the cube only rotates around its vertical axis and only one side face is white, shouldn't it have only trivial C1 (360 deg) rotational symmetry?

I'm curious: is the inspiration from the HPC-MEC circuit purely functional (based on cognitive experiments) or does this study also uses physiology-based evidence from neuroscience to inform the specifics of the architecture chosen?

From Table 1, why can we conclude that, if latent transition trajectories (z) share a more similar pattern than state transitions in HPC and MEC, then the model can extract content-independent structures from simulated environments? I can see how z does a better job at it, but the overall magnitude of the similarity appears quite low (0.235).

Have the authors conducted any experiment to help isolate with parts really contribute to the observed performance?

**Limitations:**

Yes.

**Strengths And Weaknesses:**

__Strengths:__

The text is clear and well written. The inspiration from biological HPC-MEC is original and interesting and seems well-grounded. Combining this with modern world model ideas and self-supervised learning seems highly relevant.
The model's hierarchical architecture is clean and allows for a clear interpretation of each part's capabilities.
Results were evaluated on multiple datasets, and compared against 3 SOTA models, showing evidence for robust generalization.


__Weaknesses:__

On the other hand, because the architecture is so complex, it is not clear which parts are actually essential for performance.

Although the experiments performed and datasets used are meaningful, all the dynamics used are relatively simple, namely rotations or motion involving a single object. In more complex environments, multiple objects can move independently and even interact (introducing causality, for example), which would require representations of distinct objects. Therefore the model presented seems to have limited practical application.

---

> ### Author Rebuttal · Authors · 2026-03-28
>
> We sincerely thank the reviewer for recognizing the originality of the HPC–MEC inspiration, the clarity of the hierarchical design, and the breadth of the empirical evaluation. Below, we address each question in detail.
>
> W1/Q5: Our intent was not to introduce complexity for its own sake: each major component corresponds to a specific hypothesis, and we do provide targeted ablations for them:
>
> 1. The **hierarchical HPC–MEC separation** is tested against a **unified latent space** baseline. The unified variant shows lower reuse quality and stronger texture leakage during latent transition transfer, while the full model achieves the best similarity ratio $R$, SSIM, and LPIPS in Table 2, supporting the need to separate content-rich and structure-dominant pathways.
> 2. The **CANN-based MEC dynamics** are tested by replacing them with a standard state-action concatenation mechanism (“w/o CANN”). This ablation underperforms the full model and fails on OOD latent transition transfer, showing that the path-integration mechanism is useful.
> 3. The **inverse latent-transition model** is validated directly in Appendix E: when we zero the inverse-model input, one-step prediction collapses to copying the previous frame; when we remove content binding, coarse transition structure remains but detailed generation degrades. This shows that latent transitions primarily drive the main dynamics, while content binding refines scene-specific reconstruction.
>
> W2: Thank you for this important question. We would like to clarify that the model is **trained only on SSv2**, a complex human–object interaction dataset, yet it already learns to **zero-shot extract** both simple transformations and more complex transitions in OOD robotic environments. We intentionally use clean dynamics such as rotation for analysis because they make the structure/content separation visually interpretable. Evidence that it handles **more complex dynamics** is already in the paper: (1) In Table 1 and Fig. 10, where the model extracts shared transition structure across varying contexts in OOD Franka Kitchen; (2) Fig. 5A/D/E shows latent transition reuse in these more realistic interaction settings; (3) our transfer results are stronger than baselines in Fig. 12 and Fig. 14.
>
> We do agree with the reviewer that **multiple moving objects** remain challenging. As we state in the limitations, this likely requires more explicit object-centric decomposition. Still, we believe our contribution is important because it provides a foundation for **structure extraction at the level of a primary interacting object**, which is a necessary step toward more compositional multi-object models.
>
> Q1: Thank you for your suggestions. “Abstract structures” refer specifically to **shared, reusable latent transitions** that govern how scenes evolve over time, rather than object appearance. We will add a concrete example in the introduction as you suggested.
>
> Q2: Thank you for this question. Our "1/4 period" refers to the visual similar periodicity of the latent trajectory, not the object's global symmetry. For a cube with 3 white sides, 1 brown: consecutive transitions are similar (white 1 → white 2, white 2 → white 3), so the model reuses the transition structure. By contrast, returning to white requires 180°, and transitions (white 3 → brown, brown → white 1) aren't reusable similarly. Thus, the remaining trajectory forms a larger loop. If all 4 sides were white, 1/4-period loops would overlap; if all differed, it exhibits only trivial C1 symmetry. We will clarify this in revise version.
>
> Q3: It is both. Functionally, the model follows the well-established division that HPC binds richer episodic/content information, while MEC encodes abstract relational structure. At the implementation level, the design is also physiology-informed: the MEC transition module is instantiated as a **CANN**, precisely because CANNs are the canonical computational model for **grid-cell-like** continuous attractor dynamics and velocity-driven path integration. This rationale is made explicit in Appendix A.1 and A.4. Of course, we do not claim neuron-level biological realism; rather, we abstract the core computational roles supported by the neuroscience literature.
>
> Q4: The reason is that latent transitions capture similar robotic dynamics without encoding diverse backgrounds. Table 1's primary purpose is to evaluate the degree of abstraction across representational levels in a highly challenging **OOD** setting. While the nominal tasks (e.g., "opening the upper cabinet") match, the sequences have variations in both background context and robot arm trajectories. Because extracting perfectly identical structures across such diverse variations is difficult, high absolute cosine similarities are not expected. Therefore, our key takeaway is relative rather than absolute: the latent action $z$ achieves the highest similarity among all compared representations, notably exceeding both $\Delta p$ and $\Delta g$.

---

> > ### Author Rebuttal · Reviewer_h2Ak · 2026-04-02
> >
> > Thanks for the detailed response. My concerns have been addressed and I will increase my score accordingly.

---

> > > ### Author Response · Authors · 2026-04-03
> > >
> > > Thank you again for your thoughtful follow-up and for confirming that our responses have addressed your concerns. We truly appreciate your time and the constructive feedback you have provided throughout the review process.
> > >
> > > We noticed your comment indicating an increased score, and we are very grateful for your positive reassessment. If convenient, we would appreciate it if this update could be reflected in the system as well.
> > >
> > > Thank you again for your support and engagement with our work.

---

### Decision · Program_Chairs · 2026-04-30

**Decision:**

Accept (regular)

**Comment:**

The paper presents a promising world model for structure abstraction and generalization. Three out of four reviewers provided positive ratings, while one reviewer expressed concerns regarding the strength of the neuroscience insights.

After reviewing the paper, the authors’ rebuttal, and the reviewers’ comments, I largely agree with the majority opinion that the work represents an interesting and valuable direction for world models, with a loose connection to neuroscience. The authors have also acknowledged several limitations, such as challenges in generalizing to multi-object scenes.

Reviewer CY9q raised concerns about the limited depth of neuroscience insights, which I also find valid. However, I do not view this as a critical flaw warranting rejection. That said, I encourage the authors to moderate the framing and claims related to neuroscience to better align with the actual contributions.

Overall, given the strong empirical performance and the potential of the proposed direction, the strengths outweigh the weaknesses. I therefore recommend acceptance.